# MA-LoT: Model-Collaboration Lean-based Long Chain-of-Thought Reasoning enhances Formal Theorem Proving

Ruida Wang [* 1]   Rui Pan [* 1]   Yuxin Li [* 2]   Jipeng Zhang [2]   Yizhen Jia [1]
Shizhe Diao [3]   Renjie Pi [2]   Junjie Hu [4]   Tong Zhang [1]

## Abstract

Solving mathematical problems using computer-verifiable languages like Lean has significantly impacted the mathematical and computer science communities. State-of-the-art methods utilize a single Large Language Model (LLM) to generate complete proof or perform tree search, but they fail to balance these tasks. We propose **MA-LoT**: *Model-CollAboration Lean-based Long Chain-of-Thought*, a comprehensive framework for Lean4 theorem proving to solve this issue. It separates the cognition tasks of general NL for whole-proof generation and error analysis for proof correction using the model-collaboration method. We achieve this by structured interaction of the LLM and Lean4 verifier in Long CoT. To implement the framework, we propose the novel *LoT-Transfer Learning* training-inference pipeline, which enables the Long CoT thinking capability to LLMs without special data annotation. Extensive experiment shows that our framework achieves a **61.07%** accuracy rate on the Lean4 version of the MiniF2F-Test dataset, largely outperforming DeepSeek-V3 (33.61%), single-model tree search (InternLM-Step-Prover, 50.70%), and whole-proof generation (Godel-Prover, 55.33%) baselines. Furthermore, our findings highlight the potential of combining Long CoT with formal verification for a more insightful generation in a broader perspective.

*First Authors [1]Department of Computer Science, University of Illinois Urbana-Champaign [2]Hong Kong University of Science and Technology [3]NVIDIA [4]Department of Computer Science, University of Wisconsin-Madison. Correspondence to: Ruida Wang <ruidaw@illinois.edu>.

*Proceedings of the 42$^{nd}$ International Conference on Machine Learning*, Vancouver, Canada. PMLR 267, 2025. Copyright 2025 by the author(s).

## 1. Introduction

Formal reasoning is a cornerstone of human intelligence and a key objective in machine learning (Newell & Simon, 1956), often evaluated through rigorous mathematical derivations (Yang et al., 2024a). With the rise of Large Language Models (LLMs), Chain-of-Thought (CoT) prompting has emerged to formalize reasoning by generating intermediate steps. This approach improves interpretability and enhances reasoning performance (Wei et al., 2022).

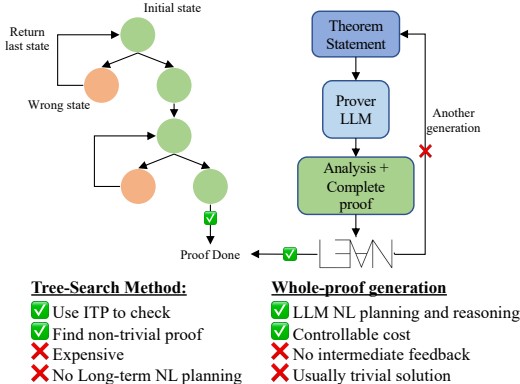

*Figure 1.* Two main directions of FL theorem proving using LLMs: Single model tree-search and whole-proof generation with their advantages/disadvantages.

However, the ambiguity of Natural Language (NL) complicates verifying intermediate steps, particularly in advanced mathematics, where there is no answer to check but theorems to prove. This challenge is exacerbated by the growing complexity of modern mathematics, which makes proof verification highly demanding and can lead to errors, as seen in the prolonged validation of Fermat's Last Theorem (Wang et al., 2024). Researchers propose grounding reasoning in rigorous first-order logic to address this, enabling automated verification via Formal Language (FL) verifiers. This framework ensures rigor and has led to the development of tools like Lean (De Moura et al., 2015; Moura & Ullrich, 2021), Isabelle (Paulson, 1994), and HOL Light (Harrison, 2009) for verifiable theorem proving.

However, writing mathematical proofs in FL requires significant expertise and effort, as most proofs involve extensive

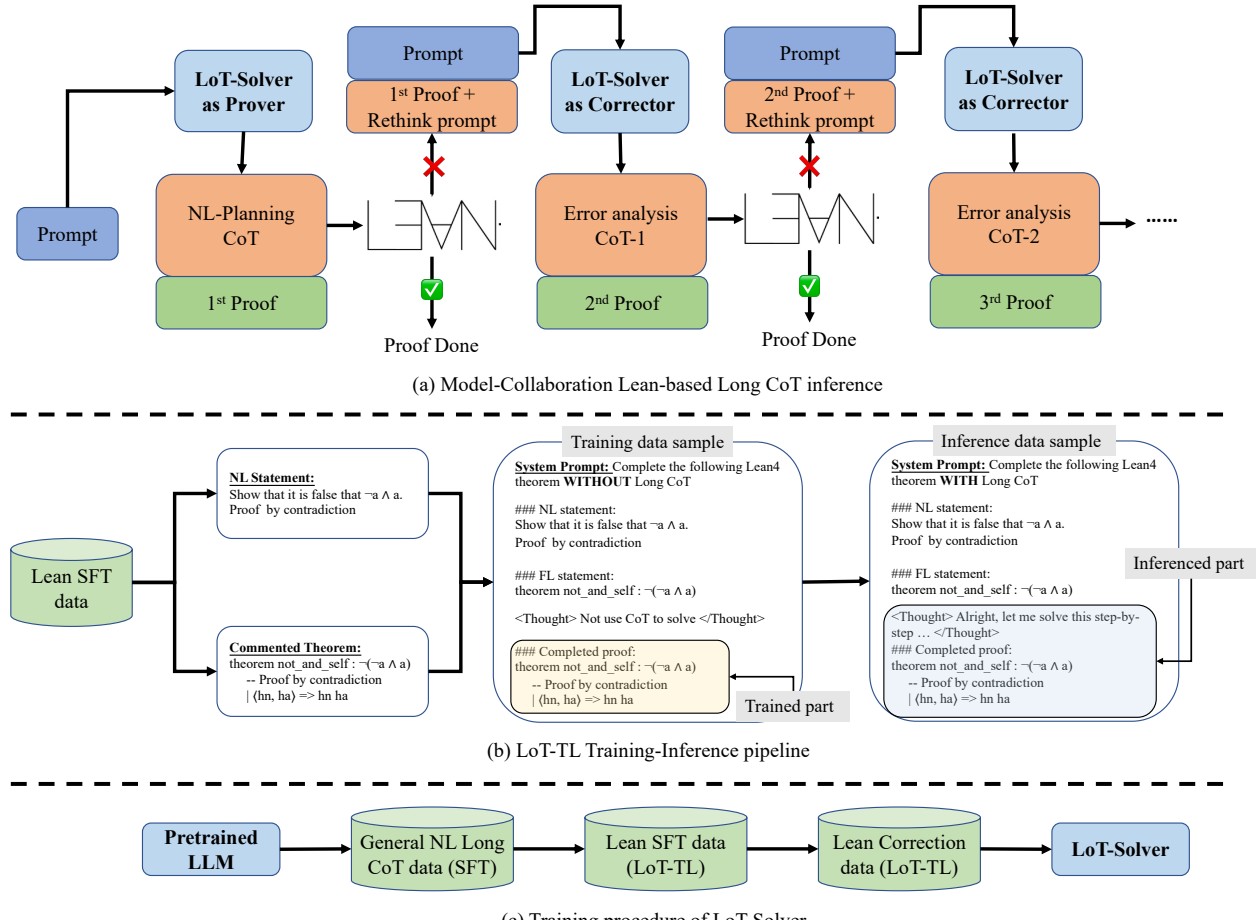

*Figure 2.* **MA-LoT Framework:** (a) Model-collaboration Lean4 theorem proving framework: The *LoT-Solver* model functions as the prover to generate initial Lean4 proofs with emergent NL planning for Lean proof in Long CoT (orange block); then it acts as corrector to analyze error from Lean executor in Long CoT to output a refined proof. (b) LoT-Transfer Learning (TL): The novel training-inference pipeline enables formal reasoning ability to emerge in Long CoT (L-CoT) without the need for specifically annotated data. This is achieved by adjusting the system prompt to control the on/off of L-CoT in training and inference. (c) Training procedure of *LoT-Solver*: We use normal SFT to train general NL L-CoT, use *LoT-TL* to train on the Lean SFT, and correction data to make Lean L-CoT an emergent capability in LLM.

repetition and application of low-resource functions (Jiang et al., 2022). With the rapid progress of LLMs, research has explored LLMs' application in FL reasoning to automate theorem proving (Polu & Sutskever, 2020; Polu et al., 2022; Jiang et al., 2021; 2022; Yang et al., 2024b; Xin et al., 2024b; Wang et al., 2024; Wu et al., 2024a; Kumarappan et al., 2024; Lin et al., 2024). Prior research follows two main approaches, namely, tree-search (Jiang et al., 2021; 2022; Lin et al., 2024; Xin et al., 2024b; Wang et al., 2024) and whole-proof generation (Polu & Sutskever, 2020; Polu et al., 2022; Yang et al., 2024b; Wu et al., 2024a; Kumarappan et al., 2024). The summary of dis/advantages of these two methods can be found in Figure 1.

Tree-search methods train an LLM to iteratively generate proof steps by predicting the next tactic based on the current proof state. This is achieved through either direct code

writing (Polu & Sutskever, 2020; Polu et al., 2022; Xin et al., 2024b; Wu et al., 2024a; Lin et al., 2024) or retrieval-based techniques (Yang et al., 2024b; Kumarappan et al., 2024). Tree-search methods apply an FL executor to verify after each generation step and can discover some non-trivial proofs. However, as proof complexity increases, tree-search methods become computationally expensive and lack high-level NL planning to control the overall structure of the proof. The lack of overall planning causes the tree-search method to be unable to find some structured proof that requires high-level NL planning to answer the question.

In contrast, whole-proof generation treats theorem proving like the code generation problem, where LLMs generate the entire proof in a single attempt using either supervised training (Wang et al., 2024; Xin et al., 2024b) or prompt engineering (Jiang et al., 2021; 2022). This approach leverages

LLMs' NL reasoning and high-level planning capabilities with predictable computation costs, but it lacks intermediate feedback from FL executors. As a result, the generated proofs often lack post-hoc analysis of errors and tend to perform badly on tedious questions that require non-trivial or tedious tricks. In summary, existing single-model approaches struggle to balance the NL reasoning with the FL verifiability, motivating the need for a more comprehensive framework.

To address the above challenges in Lean4 theorem proving, we introduce **MA-LoT**: _Model-CollAboration Lean-based Long Chain-of-Thought_, a comprehensive framework designed to coordinate multiple models for effective formal reasoning. As illustrated in Figure 2 (a), **MA-LoT** separates the cognitive process into two distinct tasks: high-level proof planning and writing, as well as fine-grained correction based on verifier feedback. This separation is enabled by a collaborative model architecture and the emergent reasoning capability of Long CoT, which was developed during training. The framework consists of two core components: a prover model, responsible for generating well-structured proofs, and a corrector model, which interprets feedback from the Lean4 verifier to refine the output. Both models are enhanced with Lean4 field-specific Long CoT capability, allowing them to reason more thoroughly before producing outputs. Moreover, we integrate Lean verification results into Long CoT to improve the system's self-reflection and correction abilities.

To implement the **MA-LoT** framework, we propose a novel training-inference pipeline called _LoT-Transfer Learning_ (LoT-TL), which is used to train the _LoT-Solver_ model, as shown in Figure 2 (b) and (c). This pipeline enables the emergence of using Long CoT to solve the FL problems without requiring specifically annotated formal data. It is achieved by leveraging transfer learning across three data sources: NL-based general Long CoT reasoning, SFT data for theorem proving under the code completion task, and in/correct proof pairs based on Lean4 feedback. Through structured adaptation, LoT-TL equips the model with awareness of FL proof states and tactics while preserving its strong NL reasoning and planning capabilities. Then, the model trained under this pipeline can generate coherent and insightful formal proofs grounded in both symbolic precision and high-level planning.

Extensive experiments demonstrate that **MA-LoT** framework effectively enhances the model's formal reasoning ability through model-collaboration design and emergent formal reasoning in Long CoT. The framework can successfully prove some of the advanced IMO and AIME problems in the MiniF2F dataset (Zheng et al., 2021), with which existing models struggle. Under similar sampling budgets, our framework achieves a 61.07% accuracy rate, surpass-

ing state-of-the-art whole-proof generation models (Godel-Prover (Lin et al., 2025), 55.33%) and tree-search baseline (InternLM-Step-Prover(Wu et al., 2024a), 50.70%).

We summarize our contributions as follows: (1) We introduce **MA-LoT**, a comprehensive model-collaboration framework to balance NL reasoning and FL verification under the Long CoT paradigm for Lean4 theorem proving. (2) We propose using Long CoT to synergically combine the nature of NL and FL, allowing the model to generate in-depth and insightful formal reasoning through NL planning and analysis. (3) We develop _LoT-TL_, a training-inference pipeline that makes field-specific Long CoT capabilities emerge to LLMs without requiring explicitly annotated datasets.

Our framework has broad potential beyond Lean4 theorem proving, demonstrating how formal verification can be effectively integrated with Long CoT reasoning. This approach reveals the potential for structured, reflective, and adaptable general text generation through iterative planning and error analysis on formal executors. To accelerate advancements in this field, we open-source our code at https://github.com/RickySkywalker/LeanOfThought-Official

## 2. Methodology

In this section, we detail the development of the **MA-LoT** framework and training procedure of _LoT-Solver_ model for Lean4 theorem proving. Our framework's core idea is to enable the model to perform Long Chain-of-Thought (CoT) reasoning under the context of Lean4 theorem proving. The Long CoT is designed to perform deep integration between Natural Language (NL) and Formal Language (FL) reasoning. Training of such Long CoT thinking model is achieved under an extreme scarcity of NL-FL aligned data faced by the entire field (Wang et al., 2024). We introduce the methods by first outlining the preliminaries of LLM formal theorem proving in Section 2.1. Then, we describe the _LoT-Transfer Learning_ (LoT-TL) training pipeline in Section 2.2. Finally, we present comprehensive details on how our trained model facilitates **MA-LoT** framework for Lean4 proof writing in Section 2.3.

### 2.1. Preliminaries

We introduce preliminary knowledge of applying Long Chain-of-Thought (CoT) LLMs to Lean4 formal theorem proving in a model-collaboration manner.

Current state-of-the-art methods treat Lean4 code as plain text and feed it directly to LLMs. There are two main branches of techniques to apply LLMs for Lean4 theorem proving. The first branch is the tree-search method (Yang et al., 2024b; Wu et al., 2024a). This method converts the

Lean4 theorem statement and current proof state (including conditions and goals based on the existing proofs) into plain text as input to LLMs and asks it to generate the next possible tactic to complete the proof. The other direction is whole-proof generation methods. It provides the LLMs with NL instruction, NL theorem statement, and Lean4 statement to the LLMs. The intended outcome is to generate a complete Lean4 proof in a single pass. This is achieved by first leveraging the LLM's NL reasoning to produce a high-level plan, then guiding the generation of the actual Lean4 proof The input-output examples for both tree-search and whole-proof generation are presented in Appendix B.

The Long CoT LLMs, represented by O1 (OpenAI, 2024), make long internal NL thinking before outputting the final answer. It largely enhances the NL math reasoning ability of LLMs through self-reflection and correction in Long CoT. However, it still struggles to provide rigorous NL proofs and typically has relatively low FL capability.

Our approach combines the strengths of tree search and whole-proof generation methods through a model-collaboration system. By leveraging Long CoT, we coordinate the interaction between NL and FL in LLMs, which allows the model to write more structured and insightful proofs.

## 2.2. LoT-TL Training Pipeline

This section introduces a simple but effective training pipeline *LoT-Transfer Learning* (LoT-TL). The pipeline intends to make LLMs have the ability to perform Lean4 field-specific Long CoT reasoning. Such ability can be obtained through LoT-TL without the need for specifically annotated Long CoT data. The main idea of LoT-TL is to leverage system prompts to regulate training and inference behaviors, which can be divided into the following stages: (1) collecting field-specific Supervised Fine-Tuning (SFT) data (Section 2.2.1), (2) training the model on general natural language Long CoT tasks (Section 2.2.2), and (3) training the model using the transfer learning method on SFT and correction data to make formal Long CoT ability emerge(Section 2.2.3). Although this paper focuses on Lean4, our framework shows potential to be applied to more fields, making LLMs obtain field-specific Long CoT capability without RL or special data annotation.

### 2.2.1. OBTAIN SFT DATA

The first step of *LoT-TL* pipeline is to gather a moderate amount of NL-FL aligned SFT data for the specific target field (in our case, Lean4). However, existing open-source datasets do not meet the requirement. They are typically small in size (e.g., MiniF2F (Zheng et al., 2021)), or omit NL annotations (e.g., DeepSeek-Prover-v1 dataset (Xin et al., 2024a)), or exhibit relatively low NL quality (e.g.,

OBT (Wang et al., 2024)), or lack Lean4 proofs (e.g., Lean-Workbook (Ying et al., 2024)).

To address this, we compile a new Lean theorem proof dataset named LoT-ProveData (LoT-PD), containing 54,465 data records. Each record contains Lean4 theorem statements, verified proofs with NL explanations as comments, and NL statements. The Lean4 theorem proofs come from two sources: the DeepSeek-Prover-v1 dataset and the annotated Lean-Workbook using TheoremLlama and DeepSeek-Prover-v1.5-RL. Next, inspired by the analysis-then-generate approach in Wang et al. (2023), we employ Qwen-2.5-72B to provide an analysis of Lean4 proofs, followed by writing the NL proof based on the analysis. Finally, we integrate these NL proofs as comments in the Lean4 code by Qwen. For data records lacking NL statements, we generate NL statements using a similar method. The core components of our LoT-ProveData are as follows:

```
{FL Statement, Commented FL proof, NL
↪   statement}
```

During proof generation for the ProverData, some incorrect proofs were also produced. We recorded these alongside their error messages to form LoT-CorrectionData (LoT-CD), consisting of 64,912 records of correct-incorrect Lean4 proof pairs with error messages of the incorrect proof. Additionally, it contains an NL statement and proof of the theorem. LoT-CD is used to train the model's error analysis and correction capability. The core part of the LoT-CD is:

```
{FL Statement, Correct FL proof, Error
↪   messages, Incorrect FL proof, NL
↪   statement}
```

These datasets work together to improve both the prover and corrector models' capability, acting as the source of strong NL-FL joint thinking ability for our model-collaboration framework.

### 2.2.2. NL LONG CoT TRAINING

In the second stage, we train a normal instruction-finetuned model to obtain the general Long CoT reasoning capability in Natural Language. We use the OpenO1-SFT-Pro dataset provided by Open-Source-O1 (2024), a 126k records dataset for general NL question-answering on math and science topics with Long CoT for training. We apply standard next-token prediction SFT, guiding the model to produce Long CoT before it outputs final answers. Throughout the NL Long CoT training, we set the system prompt as follows to explicitly instruct the model to use the Long CoT approach:

```
You are a helpful assistant who will
↪   solve every problem **WITH** Long
↪   Chain-of-Thought
```

This system prompt indicates that the model should use the Long CoT to write the answer. The training input includes the system prompt and NL question, with the expected output being the Long CoT and final answer. After training, we observe that the model gains robust NL Long CoT capabilities, which serve as a base for models to analyze and interact with Lean code. However, when applied to Lean4 reasoning, it tends to provide only NL solutions rather than outputting Lean4 code in its output section, indicating the need for further alignment.

### 2.2.3. FIELD-SPECIFIC ALIGNMENT

In the final stage of the training process of *LoT-TL* pipeline, we train the model to obtain emergent Lean4 Long CoT ability. In this training stage, we set the system prompt to instruct the model not to use Long CoT when answering. In the meantime, to make the model aware of the Long CoT structure, we use a simple placeholder to occupy the original place of the Long CoT. The above method allows us to train the model aware of the Lean4 Long CoT structure without requiring any Lean4 Long CoT data. Specifically, the system prompt is:

```
You are a helpful assistant who will
↪   solve every problem **WITHOUT**
↪   Long Chain-of-Thought
```

and the placeholder Long CoT is:

```
The user asks not to solve with Long
↪   CoT, so I will directly write the
↪   answer.
```

Under this setup, we first train on the LoT-ProveData for formal theorem proving ability, then train on the LoT-CorrectionData to make the model learn error-analysis and correction skills. The example of training data can be found in Appendix F. We also adopt the curriculum learning data sorting method from Wang et al. (2024) to stabilize training. After training, we find that the Long CoT Lean4 proving and error analysis ability emerges in the LLMs when using the system prompt to turn on Long CoT in inference. We conclude the effectiveness of the TL framework because it preserves the structure of Long CoT and enables the model to activate such capability when instructed.

Following the above steps, we train the *LoT-Solver* model based on an expert Lean4 prover with instruction fine-tuning. *LoT-Solver* is a model with Lean4 Long CoT reasoning capability, which also enables the separation of whole-proof writer and corrector.

## 2.3. Model-Collaboration Lean4 Proof Writing

This section presents the model-collaboration framework that combines the advantages of whole-proof generation and tree-search methods under the Lean-based Long CoT paradigm. We use the *LoT-Solver* as the base model for both the prover and the corrector. Under this setup, we use the prover model to write a complete proof draft (Section 2.3.1) and apply the corrector model to analyze and correct the proof based on Lean verifier feedback (Section 2.3.2).

### 2.3.1. PROVER MODEL

The prover model writes the initial Lean4 proof using a whole-proof generation strategy. Then, it is submitted to the Lean4 verifier to check its correctness and passed to the corrector model if it is wrong. We use the system prompt to turn on the Long CoT reasoning and use a specific header in Long CoT to guide the model in making a high-level proof plan. Here is the instruction template for the prover model:

```
{... **WITH** Long CoT ...}
### Instruction:
{NL statement}
{FL statement}
### Response:
Alright, I should do the following:
1. Provide the natural language
↪   analysis for the theorem based on
↪   the Natural language theorem
↪   statement.
2. Draft the Lean4 tactics I should use
↪   to solve the problem
3. Write the output Lean4 code.
```

The prover model's input and output example can be found in Appendix G. The emergent Lean reasoning ability in the Long CoT enables the model to write a better-structured proof based on its high-level plan than direct proof generation. Upon generating the proof, the prover model submits it to the Lean evaluator for verification. The theorem is then passed to the corrector model for further refinement if incorrect.

### 2.3.2. CORRECTOR MODEL

After receiving a wrong proof and Lean verifier feedback, the corrector model will systematically analyze them in Long CoT. The corrector re-evaluates and rethinks the proof strategy, then generates a revised proof that intends to correct the errors and complete the proof. In conceptual analogy, the corrector model functions similarly to the tree-search method but with greater flexibility and deeper analysis.

The instruction prompt remains identical to the prover model. We incorporate the incorrect proof and feedback from the Lean verifier in the Long CoT, followed by instructions to direct the model to analyze the error and formulate a revised proof. Detailed examples of such prompts are available in Appendix G. Then, the corrector model will

pass the new proof to the Lean4 verifier. If the new proof is still wrong, iteratively analyze the errors until success or reach the max retry limit.

The corrector model enhances theorem proving by enabling deeper reflection and systematically exploring alternative proof strategies based on error messages. This iterative correction process increases the likelihood of discovering non-trivial proofs while maintaining computational efficiency.

# 3. Experiments

We conduct comprehensive experiments on the MiniF2F-Lean4 (Zheng et al., 2021) dataset to assess the performance of the **MA-LoT** framework in formal proof writing. Specifically, we intend to prove the advantage of the **MA-LoT** framework by showing its capability to generate better-structured and more insightful proofs. We do this by showing our model has a better general performance in Section 3.3. Moreover, we perform studies on our corrector model in Section 3.4, the efficiency of the Long CoT method in Section 3.5, training components of *LoT-Solver* in Section 3.6, and a case study in Section 3.7 to further analyze the impact of individual components.

## 3.1. Experiment Setup

### 3.1.1. DATASET AND TASK

In this paper, we assess **MA-LoT**'s Lean4 reasoning capabilities on the MiniF2F-Test and Valid[1] datasets (Zheng et al., 2021; Yang et al., 2024b; Wang et al., 2024). MiniF2F is a widely used and challenging benchmark for formal theorem proving (Frieder et al., 2024), which is adopted in nearly all major studies in the field (Jiang et al., 2021; Polu et al., 2022; Jiang et al., 2022; Wu et al., 2024a; Lin et al., 2024; Yang et al., 2024b; Xin et al., 2024b; Wang et al., 2024; Azerbayev et al., 2023).

Both the test and validation datasets contain 244 Lean4 statements. The range of problems varies from high-school competition questions to undergraduate-level theorem proofs. It includes 488 problems from three sources: (1) 260 problems sampled from the MATH dataset (Hendrycks et al., 2021); (2) 160 problems from high-school math competitions, including AMC, AIME, and AMO; (3) 68 manually crafted problems at the same difficulty level as (2). Our task is to query the LLM to generate Lean4 proofs for MiniF2F problems based on their formal statements and NL descriptions. To minimize computational overhead, imports are manually configured.

### 3.1.2. BASELINES

To highlight **MA-LoT** 's capabilities, we select some of the most competitive baselines in recent years, covering both tree-search and whole-proof generation approaches. For tree-search methods, we include: Expert Iteration (Polu et al., 2022), Llemma (Azerbayev et al., 2023), ReProver (Yang et al., 2024b), Lean-STaR(Lin et al., 2024), and InternLM2.5-StepProver(Wu et al., 2024a) as our baselines. For whole-proof generation baselines, we include closed-source LLMs, represented by GPT-4-Turbo(Achiam et al., 2023), Gemini-1.5(Reid et al., 2024), and DeepSeek-V3 (Liu et al., 2024). Moreover, we include representative open-source expert models for whole-proof writing, such as DeepSeek-Math(Shao et al., 2024), TheoremLlama(Wang et al., 2024), DeepSeek-Prover-v1.5-RL (Xin et al., 2024b)[2], STP-Lean (Dong & Ma, 2025), and Godel-Prover-SFT (Lin et al., 2025).

For whole-proof generation baselines, we set the sample budget to pass@128 with 4096 context length (except for Godel-Prover, where, following the original paper, we use pass@32), balancing robustness and manageable GPU consumption. We align the search cost as closely as possible for tree-search methods to whole-proof generation.[3]

## 3.2. Implementation Details

In the model's training process, we use Openo1-SFT-Pro, LoT-ProverData, and LoT-CorrectionData to train two base models, namely DeepSeek-Prover-v1.5-SFT and Godel-Prover-SFT. For different training stages, the learning rate is as follows: 1E-5 for NL Long CoT training, 1E-7 for LoT-TL on LoT-PD, and 1E-6 for LoT-CD. The total computational cost for training is around 1 GPU day, and evaluation is 11 GPU days on a $4\times$ H100-96G cluster[4]. To evaluate our framework in detail, we present three sets of results, including: (1) **LoT** (whole-proof): pass@128 whole-proof generation result of *LoT-Solver*. (2) **MA-LoT**: Our primary evaluation result, where the prover performs 64 whole-proof generations and undergoes two rounds of corrector refinement[5]. (3) Cumulative Results: A combined evaluation aggregating all **LoT** models' outputs obtained throughout

---

[1]Although DeepSeek-Prover-v1.5 declared that they applied MiniF2F-Valid for training, yet its performance does significantly differentiate from other methods. Thus, we still keep this baseline.

[2]The reported results of DeepSeek-Prover-V1.5 and Goedel-Prover in our paper are different from the original paper because we are unable to access vLLM in our machine, and it is known that the inference without vllm will lead to slight drop of the model's capability

[3]This explains why we exclude RMaxTS for DeepSeek-Prover-v1.5, as its smallest disclosed sample budget ($1 \times 3200$) is not a comparable result to our baselines.

[4]The high evaluation cost is due to accelerated inference methods like vLLM being unable to fit our machine

[5]Because we don't need to pass a correct proof to the corrector, two rounds of correction are approximately the same as one round of whole-proof generation

| Method | Model size | Sample budget | MiniF2F-Valid | MiniF2F-Test | Average |
|---|---|---|---|---|---|
| *Tree-search Methods* | | | | | |
| ReProver (Yang et al., 2024b) | 229M | - | - | 26.5% | - |
| Llemma (Azerbayev et al., 2023) | 34B | $1 \times 32 \times 100$ | 27.9% | 25.8% | 26.85% |
| Expert Iteration (Polu et al., 2022) | 837M | $8 \times 8 \times 512$ | 41.2% | 36.6% | 38.9% |
| Lean-STaR (Lin et al., 2024) | 7B | $64 \times 1 \times 50$ | - | 46.3% | - |
| InternLM2.5-StepProver (Wu et al., 2024a) | 7B | $2 \times 32 \times 600$ | 56.0% | 50.7% | 53.35% |
| *Whole-proof generation* | | | | | |
| GPT-4-Turbo (Achiam et al., 2023) | > 1T | | 25.41% | 22.95% | 24.18% |
| DeepSeek-Math (Shao et al., 2024) | 7B | | 25.80% | 24.60% | 25.20% |
| Gemini-1.5-pro (Reid et al., 2024) | - | pass@128 | 29.92% | 27.87% | 28.90% |
| TheoremLlama (Wang et al., 2024) | 8B | | 38.52% | 35.66% | 37.66% |
| DeepSeek-Prover-v1.5-RL (Xin et al., 2024b) | 7B | | 54.10% | 48.36% | 51.23% |
| STP-Lean (Dong & Ma, 2025) | 7B | | - | 56.15% | - |
| Godel-Prover (Lin et al., 2025) | 7B | pass@32 | - | 55.33% | - |
| DeepSeek-V3 (Liu et al., 2024) | 685B | pass@32 | - | 33.61% | - |
| *Ours* | | | | | |
| DeepSeek-Prover + LoT (whole-proof) | | pass@128 | 62.70% | 52.05% | 57.42% |
| DeepSeek-Prover + MA-LoT | | $64 + 32 \times 2$ | **64.34%** | 54.51% | **59.22%** |
| Godel-Prover + LoT (whole-proof) | 7B | pass@32 | - | 57.79% | - |
| Godel-Prover + MA-LoT | | $16 + 8 \times 2$ | - | **61.07%** | - |
| MA-LoT | | cumulative | **65.98%** | **63.93%** | **64.96%** |

*Table 1.* Main experimental results of **MA-LoT**. Our results are presented as *Base model + method*. The LoT (whole-proof) indicates using the whole-proof results of our *LoT-Solver* model, and **MA-LoT** indicates the result of our whole pipeline. The sampling budget $x + k \times y$ in the MA-LoT framework indicates we first perform $x$ whole-proof writing using the prover model and $k$ rounds of correction using the corrector model. In practice, one round of correction costs approximately half of the whole-proof generation budget, resulting in $y = \frac{1}{2}x$.

| Method | Prover | round 1 | round 2 | round 3 |
|---|---|---|---|---|
| DS-Prover-v1.5 | 51.64% | 53.28% | 54.51% | 55.33% |
| Godel-Prover-SFT | 54.92% | 59.43% | 61.07% | 61.89% |
| Prover as Corrector | 54.92% | 56.15% | 57.38% | - |

*Table 2.* Results from multiple rounds of correction using DeepSeek-Prover and Godel-Prover as base models, along with the outcomes of using the prover to perform correction.

| Method | MiniF2F-Test |
|---|---|
| DeepSeek-Prover-v1.5-SFT (base model) | 46.31% |
| LoT-Solver witch-off Long CoT | 49.18% |
| w/o Long CoT training (on RL model) | 48.36% |
| base model + Long CoT | 46.72% |
| base model + Long CoT + SFT | 50.00% |
| LoT-Solver | **51.64%** |

*Table 3.* Ablation study result in pass@64 under DeepSeek-Prover as base model.

| Method | Godel-Prover-SFT | MA-LoT-Godel |
|---|---|---|
| Sample budget | pass@32 | $16 + 2 \times 8$ |
| Avg. tokens gen | 492.10 | 657.54 |
| MiniF2F-Test | 58.20% | **61.07%** |

*Table 4.* Long CoT efficiency study result.

the experiment process.

### 3.3. Results

Table 1 presents our main results, showing that **MA-LoT** achieves **61.07%** accuracy rate on MiniF2F-Test benchmark and 57.79% for *LoT-Solver* using whole-proof generation. After the enhancement of MA-LoT, we achieve 10.37% on Godel-Prover and 12.72% on DeepSeek-Prover on the MiniF2F-Test benchmark. This significant and uniform improvement of our framework indicates its effectiveness. Detailed analysis also shows that our models can solve IMO and AIME problems that previous models struggled with. Our model surpasses state-of-the-art tree-search (InternLM-2.5) and whole-proof generation (Godel-Prover) baselines, demonstrating that our proposed model-collaboration framework based on Long CoT excels in formal theorem proving.

**MA-LoT** outperforms all tree-search baselines by at least 20.45% because its prover model constructs proofs with high-level NL planning using emergent Lean Long CoT reasoning capability. This indicates our prover model can leverage LLMs' strong NL reasoning ability in its Long CoT, which leads to more comprehensive proofs. Additionally, **MA-LoT** surpasses whole-proof generation baselines by at least 10.37%, as its corrector model analyzes, reflects, and reformulates proofs based on Lean4 executor feedback in Long CoT. The strong performance also demonstrates the effectiveness of our ideas of integrating FL verification in NL

Long CoT reasoning with its emergent capability. Notably, the DeepSeek-Prover + MA-LoT is based on DeepSeek-Prover-v1.5-SFT; it outperforms its RL-trained variant by 6.15%. This suggests that our model-collaboration framework and Lean-based Long CoT methodology align more naturally with formal theorem proving than RL alone.

The comparison between **LoT** (whole-proof) and **MA-LoT** further highlights the importance of our model-collaboration framework. We observe a 2.46% improvement by reallocating the computation resources from the prover model, which performs more whole-proof generation, to the corrector model, which analyzes and refines proofs based on Lean executor feedback. This validates the necessity of a model-collaboration system over relying solely on a prover model. In summary, these results confirm that Long CoT reasoning, combined with formal verification and a model-collaboration paradigm, enhances the discovery of non-trivial and in-depth proofs, thereby validating the effectiveness of our proposed method.

### 3.4. Corrector study

To assess the impact of the correct model in **MA-LoT**, we present the cumulative accuracy on the MiniF2F-Test across different rounds of correction in Table 2 for two of our base models. The prover column presents the pass@64 (or 16) accuracy rate of the prover model in whole-proof generation, while Round-$i$ columns indicate successive correction rounds. The results indicate that during the first three correction rounds, the corrector model successfully corrects an average of 11.55% theorems that the prover cannot answer. Our analysis shows that most corrected proofs belong to IMO, AIME, and high-level MATH problems, which are particularly difficult for prior models. Such additional solved problems highlight the corrector's ability to analyze feedback from the Lean4 executor feedback using emergent Lean capability in Long CoT to discover non-trivial proofs. The case study for the analysis and regeneration of new proofs can be found in Section 3.7 and Appendix E.

Additionally, to validate the effectiveness of the expert corrector, we conducted an experiment that used prover to perform corrections. In particular, we apply the *LoT-Solver* based on Goedel-Prover-SFT. We add the wrong Lean4 proof and error message as a comment in the Lean4 code block and ask the prover to correct it. Results are shown in the last row of Table 2. It shows a clear drop in performance in both correction rounds. This drop is because the prover model tends to directly repeat the wrong proof instead of using the Long CoT to analyze the error and correct the proof. This experiment shows the importance of an expert corrector model, further validating the effectiveness of our idea of using FL as a backbone for Long CoT in NL.

### 3.5. Long CoT Efficiency Study

To further study the property of Long CoT, we perform this experiment to test whether **MA-LoT** is able to search for the answer more efficiently. In this experiment, we measure the GPU hours as well as the tokens generated in both **MA-LoT** and the base model. We observed that the **MA-LoT** generated 33% additional tokens and costs 70% additional GPU hours compared to the base model. Thus, to make a fair comparison in the sense of computation hours, we enlarge the base model's sample budget to 1.7 times in this experiment; the results are presented in the Tab. 4. The table shows that despite having a smaller sampling budget, **MA-LoT** still outperforms the base model by 4.93%, which indicates that **MA-LoT** is a better way to allocate computation power.

### 3.6. Ablation Study

To show the effectiveness of each component of our *LoT-TL* training pipeline, we conduct this thorough ablation study. We demonstrate that the elements in *LoT-TL* pipeline work synergistically to strengthen the model's formal theorem-proving capability through integrating FL in Long CoT. We use DeepSeek-Prover-v1.5-SFT as our base model and apply the pass@64 accuracy rate on the whole-proof generation method for this set of experiments. The results are presented in Table 3.

#### 3.6.1. EFFECT OF TRAINING STAGES

We evaluate training progression by measuring performance across key intermediate models, namely *base model*, *base model + Long CoT*, and *base model + Long CoT + SFT*, as shown in Table 3. Results indicate that training solely on NL CoT data provides minimal improvement. It suggests that the model lacks field-specific information to finish a comprehensive Lean4 proof under pure general NL Long CoT training. However, incorporating SFT data with the *LoT-TL* training method yields a marked improvement, demonstrating the effectiveness of transfer learning in equipping models with Lean4 Long CoT capabilities. Interestingly, although it is not designed for whole-proof writing, additional training with correction data further enhances the performance. This improvement likely arises from the model's development of self-analysis capabilities in Lean4 code, allowing it to avoid potentially wrong solutions.

#### 3.6.2. SWITCH-OFF LONG CoT

This experiment shows that the strong FL reasoning power of *LoT-Solver* comes from the emergent formal reasoning ability in Long CoT, rather than trivially stacking more data. It uses our *LoT-Solver* model to write Lean4 proofs directly using the code completion method without Long CoT. We find the performance drop from 51.64% to 49.18%

because the model does not take an explicit high-level plan in Long CoT, making it unable to finish some questions in the induction field.

### 3.6.3. ABLATION OF LONG COT

To validate the quality of integration between NL and FL in Long CoT, we fine-tune the DeepSeek-Prover-v1.5-RL model directly using our SFT dataset without Long CoT reasoning. We can find that the performance of w/o Long CoT model (48.36%) is lower than *LoT-Solver* (51.64%). The results confirm that Long CoT plays a crucial role in Lean4 theorem proving, offering structured reasoning that outperforms a direct RL-based additional data fine-tuned model.

### 3.7. Case Study

This section presents the general results of case studies of the **MA-LoT** framework. Due to the limited space, we leave detailed examples in Appendix E. From the examples, we can observe a clear collaboration between models. The prover can use a high-level plan to prove advanced MATH theorems, and the corrector can analyze the feedback from the Lean executor to formulate a correct proof of an IMO-level problem. The content in Long CoT also demonstrates the formal reasoning abilities that emerge from our *LoT-TL* training pipeline in both prover and corrector. These observations qualitatively validate the model-collaboration system's design and emergence of formal reasoning ability in Long CoT, demonstrating its ability to combine high-level planning with iterative refinement in Lean-based Long CoT.

## 4. Related Work

### 4.1. Lean4 Theorem Proving using LLMs

The application of LLMs for FL proving has been a hot topic for study in recent years. The tree-search including works represented by Expert Iteration (Polu et al., 2022), Re-Prover (Yang et al., 2024b), Lean-Star (Lin et al., 2024), and InternLM-Step-Prover (Wu et al., 2024b). This direction does not take full consideration of the LLMs' NL reasoning ability and costs exponentially increasing computation power. Another direction treats formal languages as code and asks the LLMs to do the whole-proof generation without interaction with the Lean executor to fully use the NL reasoning ability of LLMs. Significant works includes DeepSeek-Prover (Xin et al., 2024b;a), TheoremLlama (Wang et al., 2024), and Llemma (Azerbayev et al., 2023). Works in this direction tend to overlook the verification signals from Lean executors or do not thoroughly think about the error messages. Although there is some early work (First et al., 2023; Zheng et al., 2023) that tries to combine both models, it did not perform well because of the lack of thorough NL

reasoning and multi-round thinking in Long CoT.

### 4.2. LLM Collaboration

Traditional RL methods offer a training method solution for general reasoning and decision-making processes, but often suffer from low sample efficiency and generalization problems (Pourchot & Sigaud, 2018). With the fast-developing reasoning and instruction-following ability of LLMs, many researchers began to separate the cognition task for a single LLM (Wang et al., 2022; Ma et al., 2024). The primary method for the separation of tasks is to design special prompts and in-context examples to let LLMs interact with the outsourcing tools using actionable responses (Xie et al., 2023; Yang et al., 2023). Further efforts were made to apply specialized training to enhance their field-specific capabilities (Xu et al., 2023; Reed et al., 2022). In the context of formal reasoning, most tree-search methods (Yang et al., 2024b; Lin et al., 2024; Wu et al., 2024a) apply an LLM to continuously query the executor and receive feedback to further refine the proof. Because of the huge difference between FL and NL, such methods are unable to provide a high-level analysis of the problem and provide a structured response.

## 5. Conclusion

This paper introduces **MA-LoT**, a comprehensive model-collaboration framework for formal theorem proving that leverages Lean-based Long Chain-of-Thought (CoT) reasoning. **MA-LoT** separate high-level proof planning from fine-grained correction, overcoming limitations of single-model approaches that either fail to harness LLMs' NL reasoning capability or lack integration with formal feedback. By coordinating a prover and a corrector model through Long CoT, **MA-LoT** produces more structured, coherent, and insightful proofs. To support this framework, we introduce *LoT-TL* training-inference pipeline. This pipeline enables LLMs to develop domain-specific Long CoT reasoning capability through transfer learning without requiring specifically annotated Lean Long CoT data. We conduct extensive experiments on the MiniF2F benchmark, where the **MA-LoT** achieves 61.07% accuracy, surpassing all baselines, including both tree-search and whole-proof generation methods. These results highlight the value of combining formal verification with structured and iterative Long CoT reasoning. Beyond theorem proving, *LoT-TL* we proposed offers a general method for enabling Long CoT reasoning in other specialized domains without expensive RL or data annotation. Additionally, the success of the model-collaboration Long CoT in Lean4 suggests broader applications of formal verification for enhancing structured reasoning across diverse fields.

## Impact statement

This paper presents work whose goal is to advance the formal theorem proving using LLMs. The potential social impact is majorly in the field of education. With the increasing number of formal languages used in graduate-level education, a more advanced formal theorem proving model may result in educators being unable to distinguish the model-generated results and student writing results. Despite the societal consequences of improving formal reasoning systems, specific discussions of ethical concerns are still too early at this stage.

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

## A. Term chart

To make the reader better understand the terms, we provide this chart that explains every term, abbreviation, and corresponding tool in detail.

1. **NL** (Natural Language): Refers to language that humans use in our daily life, often unable to perform auto-verification.

2. **FL** (Formal Language): A structured and mathematically precise representation of logic and proofs, which ensures rigorous verification and eliminates ambiguities present in NL reasoning.

3. **Lean4**: A functional programming language and interactive theorem prover developed for formalizing mathematics and verifying proofs.

4. **Lean Executor**: The built-in proof verification engine of Lean4. It evaluates proof steps, checks for correctness, and ensures that every logical inference follows strict formal verification rules.

5. **Long CoT (Long Chain-of-Thought)**: The reasoning structure provided by OpenAI-O1 (OpenAI, 2024) that performs long and detailed thinking before making the final output. Different from standard CoT, Long CoT allows for multi-step logical reasoning before proof generation, reflection, and iterative refinement from self-check of Lean4 feedback.

6. **MA-LoT** (Model-CollAboration Lean-based Long Chain-of-Thought framework): Our proposed model-collaboration framework for formal theorem proving.

7. **LoT-TL** (LoT-Transfer Learning): The transfer learning pipeline we propose to enable LLMs with Lean4 Long CoT capability without the need for a specially annotated dataset.

8. **LoT-Solver**: The model we train through *LoT-TL* pipeline that serves both as prover and corrector. It is a high-standard Lean4 theorem proving model with Long CoT capability to control the type of cognition task and enhance formal thinking.

9. **LoT-PD & LoT-CD** (LoT-ProveData) & (LoT-CorrectionData): The LoT-PD is a dataset containing verified Lean4 theorem proofs and NL statement and proof. Its main function is to provide basic capability for the prover. The LoT-CD is a dataset containing incorrect error messages and correct Lean4 proofs together with NL annotations. It mainly provides the capability for the prover model to refine proofs.

| Training Steps | MiniF2F-Test |
|---|---|
| **125 Steps** | 32.79% |
| **250 Steps** | 35.25% |
| **1,250 Steps** | 45.08% |
| **2,500 Steps** | 47.54% |
| **13,616 Steps (full training)** | 51.64% |

*Table 5.* MiniF2F-Test result on different data-scale trained models

## B. Examples for tree-search and whole-proof generation

Following Section 2.1, we present the input-output example for tree-search method and whole-proof generation method here.

Example for whole-proof generation:

```
### Input Example
Complete the following Lean 4 code:
theorem algebra_sqineq_unitcircatbpamblt1
  (a b: ℝ)
  (h₀ : a^2 + b^2 = 1) :
  a * b + (a - b) ≤ 1 := by
### Output Example
  -- We have that (a - b - 1)^2 ≥ 0.
  have h₁ : 0 ≤ (a - b - 1) ^ 2 :=
    sq_nonneg _
  -- By expanding, we have:
  -- 0 ≤ a^2 -ab-a-ab+b^2 +b-a+b+1.
  linarith [h₀, sub_add_cancel a b]
```

Example for tree-search method:

```
### Input Example
DECL MyNat.mul_pow
GOAL a b n : N
⊢ (a * b) ^ n = a ^ n * b ^ n
### Output Example
PROOFSTEP induction n with t Ht
```

## C. Scaling Law Study

We conduct an additional experiment to investigate whether our model adheres to scaling laws by evaluating its performance across varying training dataset scales. Specifically, we train the model with four distinct step sizes (125, 250, 1250, and 2500 steps) under a fixed batch size of 4 and evaluate its accuracy on the MiniF2F-Test benchmark using the Long Chain-of-Thought (CoT) format. The resulting accuracy rates are presented in Table 5. To analyze these results, we plot the relationship between log-scaled training steps and model accuracy in Figure 3. A linear approximation of the data yields a strong coefficient of determination ($R^2 = 0.9664$), indicating a near-linear improvement in performance as training steps increase logarithmically. This aligns with the predictions of scaling laws, suggesting that our approach benefits significantly from larger-scale train-

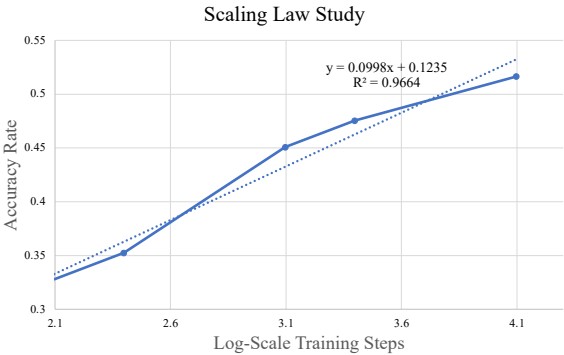

*Figure 3.* Log-scale training steps and accuracy rate on MiniF2F of trained model. The dashed line is a linear regression approximation.

ing. Our findings further highlight the potential for enhanced performance with expanded resources to annotate additional Lean4 data.

## D. ProofNet result

We perform a small-scale baseline comparison on the ProofNet benchmark using the Goedel-Prover-SFT and MA-LoT(Goedel), where **MA-LoT** achieves 15.47% of accuracy rate under $16+16$ metric while the original Goedel-Prover is 12.15% under pass@32. The relatively low performance of both models is that Goedel-Prover does not fit the ProofNet well as reported in Lin et al. (2025). However, we can still see the improvement of our method compared to the baseline, indicating **MA-LoT**'s effectiveness is not constrained to the baseline.

## E. Case studies

We present two case studies in Appendix E.1 and E.2. We can see that the model-collaboration framework can write a more coherent and well-structured proof by making a high-level NL plan and executing such a plan in the prover. The corrector can analyze the unsolved goals, make plans for correction, and write correct proof. This is achieved by our proposed training procedure.

### E.1. Prover analysis

This is an example of a prover-generated result, in which we can see that the prover model learns to write a high-level proof plan in the Long CoT and details execute the plan when writing the Lean4 code

```
### Provided Long CoT Header
<Thought>
Alright, I should do the following:
  1. Provide the natural language analysis for the theorem based on the Natural
  ↪  language theorem statement.
  2. Draft the Lean4 tactics I should use to solve the problem
  3. Write the output Lean4 code.
The user also asks that I should avoid using the keyword `sorry` to give up the
↪  proof, so I will not write it in my Lean4 code.
### Generated results:
The `mathd_numbertheory_458` can be proofed by leveraging the properties of
↪  modular arithmetic. Specifically, we know that if `n % 8 = 7`, then `n` can
↪  be expressed as `n = 8k + 7` for some integer `k`. We need to find `n % 4`.
First, observe that `n % 8 = 7` implies `n = 8k + 7`. Since `8k` is divisible by
↪  `4`, we only need to consider the remainder when `7` is divided by `4`, which
↪  is `3`. Therefore, `n % 4 = 3`.
Now, translating this reasoning into Lean4 code, we can use the properties of
↪  modular arithmetic to directly derive the result.
</Thought>
<Output>
```lean4
theorem mathd_numbertheory_458 (n : ℕ) (h₀ : n % 8 = 7) : n % 4 = 3 := by
  -- We know that n leaves a remainder of 7 when divided by 8.
  have h₁ : n % 8 = 7 := h₀
  -- We need to find the remainder when n is divided by 4.
  have h₂ : n % 4 = 3 := by
    -- Using the fact that 4 divides 8, we can use the modulo operation
    ↪  properties.
    rw [← Nat.mod_mod_of_dvd n (by decide : 4 | 8)]
    -- Simplify the expression using the given remainder when divided by 8.
    simp [h₁]
  -- Therefore, the remainder when n is divided by 4 is 3.
  exact h₂
```&
</Output>
```

### E.2. Corrector analysis

This is an example of error correction that successfully solves an IMO problem, We can see that the model first analyzes the error message, then makes a proof plan, and then writes a correct proof for another IMO-level problem

```
### Provided Long CoT Header
<Thought>
Alright, I need to prove the theorem prover_data_68493 using the Lean4 code. Here
↪  is my draft of the proof:

```lean4
theorem imo_1983_p6
  (a b c : ℝ)
  (h₀ : 0 < a ∧ 0 < b ∧ 0 < c)
```

```
  (h₁ : c < a + b)
  (h₂ : b < a + c)
  (h₃ : a < b + c) :
  0 ≤ a^2 * b * (a - b) + b^2 * c * (b - c) + c^2 * a * (c - a) := by
  /-
  To prove that \( a^2 b(a - b) + b^2 c(b - c) + c^2 a(c - a) \geq 0 \) for
  ↪  positive real numbers \( a \), \( b \), and \( c \) satisfying certain
  ↪  conditions related to being part of a triangle, we can proceed as follows:
  1. **Nonlinear Arithmetic**: Utilize nonlinear arithmetic (`nlinarith`) to
  ↪  verify inequalities involving squares and products of differences between
  ↪  variables. These operations inherently handle cases where expressions might
  ↪  not hold due to cancellation or sign changes.
  Equality holds if all pairwise differences among \( a \), \( b \), and \( c \)
  ↪  become zero simultaneously. However, since they must satisfy strict
  ↪  positivity constraints imposed by `triangle`, such an eventuality cannot
  ↪  occur under normal circumstances unless specified otherwise. Nonetheless,
  ↪  within the confines of algebraic manipulation without additional
  ↪  constraints beyond those provided, no immediate contradictions arise
  ↪  regarding potential zeros leading to equality.
  -/
  -- Introduce intermediate hypotheses ensuring divisibility properties do not
  ↪  disrupt expected behavior.
  have ha' : True := trivial
  have hb' : True := trivial
  have hc' : True := trivial
  -- Employ nonlinear arithmetic to iteratively check for inequalities involving
  ↪  sums and products of our main variables and constants.
  repeat' nlinarith [mul_pos h₀.1 h₀.2.1, mul_pos h₀.2.1 h₀.2.2, mul_pos h₀.2.2
  ↪  h₀.1,
    sub_self a, sub_self b, sub_self c, add_sub_cancel'_right a b,
    ↪  add_sub_cancel'_right b c,
    add_sub_cancel'_right c a, mul_nonneg (ha'.le) (hb'.le), mul_nonneg (hb'.le)
    ↪  (hc'.le),
    mul_nonneg (hc'.le) (ha'.le)]
```&
Let me test it in Lean4
Emmm, it seems the above proof is wrong.
Let me check the error messages.
OK, Here is the error messages:
```bash
line 13

unsolved goals
a b c : ℝ
h₀ : 0 < a ∧ 0 < b ∧ 0 < c
h₁ : c < a + b
h₂ : b < a + c
h₃ : a < b + c
ha' hb' hc' : True
⊢ 0 ≤ a ^ 2 * b * (a - b) + b ^ 2 * c * (b - c) + c ^ 2 * a * (c - a)
```&
So, I will rethink a Lean4 proof following the steps

  1. Provide the natural language analysis for the theorem based on the Natural
  ↪  language theorem statement, Lean4 theorem statement, my previous proof and
  ↪  the error message.
  2. Draft the Lean4 tactics I should use to solve the problem
  3. Write the output Lean4 code.
Let me analysis the wrong Lean4 solution through the error messages.
### Generated results"
The error message indicates that the proof is incomplete.
The error message says:
```

unsolved goals
a b c : ℝ
$h_0$ : 0 < a ∧ 0 < b ∧ 0 < c
$h_1$ : c < a + b
$h_2$ : b < a + c
$h_3$ : a < b + c
ha' hb' hc' : True
⊢ 0 ≤ a ^ 2 * b * (a − b) + b ^ 2 * c * (b − c) + c ^ 2 * a * (c − a)
```&
So, the main issue is that the proof is incomplete, and the final goal `0 ≤ a^2 *
↪  b * (a − b) + b^2 * c * (b − c) + c^2 * a * (c − a)` is not being addressed
↪  properly.
Therefore, I need to construct a proper proof strategy that addresses all the
↪  conditions and goals properly.
First, I need to understand the mathematical structure of the inequality `a^2 * b
↪  * (a − b) + b^2 * c * (b − c) + c^2 * a * (c − a)`.
This expression is symmetric in `a`, `b`, and `c`.
Given that `a`, `b`, and `c` are positive real numbers and they satisfy certain
↪  conditions related to being part of a triangle (triangle inequalities), I can
↪  think about the properties of such expressions.
One approach is to consider the symmetry and positivity of `a`, `b`, and `c`.
Since `a`, `b`, and `c` are sides of a triangle, they satisfy the triangle
↪  inequalities:
- `a + b > c`
- `b + c > a`
- `c + a > b`
Additionally, since `a`, `b`, and `c` are positive, we can use properties of real
↪  numbers and inequalities to prove the desired result.
Another approach is to use algebraic manipulation or inequalities like AM−GM
↪  inequality or Cauchy−Schwarz inequality.
Given the complexity of the expression, perhaps rearranging terms or applying
↪  known inequalities could help.
Also, considering the conditions of the triangle, perhaps expressing the
↪  variables in terms of each other or using substitution could simplify the
↪  problem.
I need to ensure that the final proof is logically sound and does not rely on
↪  assumptions that are not justified.
Therefore, I will proceed to construct a revised proof based on these
↪  considerations.
Let me proceed with the thought process and output the corrected Lean4 code.
</Thought>

<Output>
```lean4

```
/-- Let $a$, $b$, and $c$ be the lengths of the sides of a triangle. Prove that
$a^2 b(a - b) + b^2 c(b - c) + c^2 a(c - a) \geq 0$.
-/
theorem imo_1983_p6
  (a b c : ℝ)
  (h₀ : 0 < a ∧ 0 < b ∧ 0 < c)
  (h₁ : c < a + b)
  (h₂ : b < a + c)
  (h₃ : a < b + c) :
  0 ≤ a^2 * b * (a - b) + b^2 * c * (b - c) + c^2 * a * (c - a) := by
  -- We will use the fact that the square of any real number is non-negative.
  have h₄ : 0 ≤ (a - b)^2 := sq_nonneg (a - b)
  have h₅ : 0 ≤ (b - c)^2 := sq_nonneg (b - c)
  have h₆ : 0 ≤ (c - a)^2 := sq_nonneg (c - a)
  -- Using these inequalities, we can conclude the desired inequality using
  ↪ non-linear arithmetic.
  nlinarith [mul_pos h₀.1 h₀.2.1, mul_pos h₀.2.1 h₀.2.2, mul_pos h₀.2.2 h₀.1,
    mul_pos (sub_pos.mpr h₁) (sub_pos.mpr h₂), mul_pos (sub_pos.mpr h₂)
    ↪ (sub_pos.mpr h₃),
    mul_pos (sub_pos.mpr h₃) (sub_pos.mpr h₁)]
```&
</Output>
```

Another example is demonstrated as follows:

```
@ Natural language theorem statement:
mathd_algebra_419:
What is the value of $-a-b^2+3ab$ if $a=-1$ and $b=5$? Show that it is -39.

@ Lean4 theorem statement:
```lean4
theorem mathd_algebra_419
  (a b : ℝ)
  (h₀ : a = -1)
  (h₁ : b = 5) :
  -a - b^2 + 3 * (a * b) = -39 := by
```&

@ Lean4 theorem statement and proof with explanatory comments preceding each line
↪ of code:
### Response:
<Thought>
Okay the user asks me to write Lean4 theorem proof using Long CoT, I should do
↪ the following:

  1. Provide the natural language analysis for the theorem based on the Natural
  ↪ language theorem statement.

  2. Draft the Lean4 tactics I should use to solve the problem

  3. Write the output Lean4 code.
```

The user also asks that I should avoid using the keyword `sorry` to give up the
↪   proof, so I will not write it in my Lean4 code.

The natural language analysis for the theorem `mathd_algebra_419` is:

Given a = -1 and b = 5, we need to compute the value of the expression -a - b² +
↪   3ab and show that it equals -39.

The steps to solve this problem are:

1. Substitute a = -1 and b = 5 into the expression.
2. Simplify the expression step by step.
3. Verify that the result is -39.

The Lean4 code for this theorem would involve substituting the given values for a
↪   and b into the expression and then simplifying it using algebraic operations.

Now, let's outline the Lean4 code:

The theorem statement is:

theorem mathd_algebra_419
  (a b : ℝ)
  (h₀ : a = -1)
  (h₁ : b = 5) :
  -a - b^2 + 3 * (a * b) = -39 := by

The proof would involve substituting the values of a and b into the expression
↪   and simplifying:

First, substitute a = -1 and b = 5 into the expression:

-a - b² + 3ab becomes:

-(-1) - (5)² + 3*(-1)*(5)

Simplify each term:

-(-1) = 1

-(5)² = -25

3*(-1)*(5) = -15

So the expression becomes:

1 - 25 - 15

Now compute the result:

1 - 25 = -24

-24 -15 = -39

Thus, the value of the expression is $-39$.

Now, let's write the Lean4 code.

The code would start with the theorem statement, then use the `rw` tactic to
↪ substitute the values of a and b.

Wait, but in Lean4, we can directly substitute the values into the expression.

So the code would be:

```
theorem mathd_algebra_419
  (a b : ℝ)
  (h₀ : a = -1)
  (h₁ : b = 5) :
  -a - b^2 + 3 * (a * b) = -39 := by

  rw [h₀, h₁]
  ring
```

But wait, let me check. The expression is $-a - b^2 + 3ab$. So substituting $a = -1$
↪ and b = 5:

$-(-1) - 5^2 + 3*(-1)*5 = 1 - 25 - 15 = -39$.

Yes, that's correct.

So the code can be written as:

```
rw [h₀, h₁]
```

Then use the `ring` tactic to simplify the expression.

So the complete code would be:

```
theorem mathd_algebra_419
  (a b : ℝ)
  (h₀ : a = -1)
  (h₁ : b = 5) :
  -a - b^2 + 3 * (a * b) = -39 := by

  rw [h₀, h₁]
  ring
```

Alternatively, using the `simp` tactic with the appropriate lemmas.

But in this case, the `ring` tactic is sufficient.

So the code is straightforward.

I should test this code in Lean4 to ensure that it works.

But since the user asked to avoid using `sorry`, I will not write the code.
↪ Instead, I will provide the explanation.

```
</Thought>

<Output>

To solve the problem, we need to evaluate the expression \(-a - b^2 + 3ab\) given
↪  \(a = -1\) and \(b = 5\), and show that the result is \(-39\).

### Key steps:
1. **Substitute the given values into the expression:**
   - Replace \(a\) with \(-1\) and \(b\) with \(5\).
   - The expression becomes \(-(-1) - (5)^2 + 3(-1)(5)\).

2. **Simplify each term:**
   - \(-(-1) = 1\)
   - \((5)^2 = 25\)
   - \(3(-1)(5) = -15\)

3. **Combine the simplified terms:**
   - \(1 - 25 - 15 = -39\)

### Proof:
```lean4
theorem mathd_algebra_419
  (a b : ℝ)
  (h₀ : a = -1)
  (h₁ : b = 5) :
  -a - b^2 + 3 * (a * b) = -39 := by
  rw [h₀, h₁]
  ring
```&

### Explanation:
- **Substitution:** The `rw [h₀, h₁]` tactic replaces \(a\) with \(-1\) and \(b\)
↪  with \(5\) in the expression.
- **Simplification:** The `ring` tactic simplifies the expression using algebraic
↪  rules, resulting in \(-39\).

This approach ensures that we efficiently and correctly evaluate the expression
↪  by substituting the given values and simplifying step-by-step. The final
↪  result confirms that the value of the expression is indeed \(-39\).
</Output>
```

From the example above, we can clearly see the saying of *wait*, which indicates a rethinking process of the Long CoT. Such rethinking can help the model to reach conclusion that needs more thorough thinking.

### E.3. Theorem only solved by MA-LoT

We have observed that algebra_2varlineareq_fp3zeq11_3tfm1m5zeqn68_feqn10_zeq7 is only solved in the MA-LoT setup. From the original code draft and the error analysis, we can spot that our corrector framework is able to restructure the proof from the previous left and right cases into direct solving. This indicates our method's superiority in rethinking based on formal guidance.

```
<Thought>
Alright, I need to prove the theorem lean_workbook_plus_68493 using the Lean4
↪  code. Here is my draft of the proof:
```

```lean4
import Mathlib
import Aesop

set_option maxHeartbeats 0

open BigOperators Real Nat Topology Rat
theorem algebra_2varlineareq_fp3zeq11_3tfm1m5zeqn68_feqn10_zeq7
  (f z : ℂ)
  (h₀ : f + 3*z = 11)
  (h₁ : 3*(f - 1) - 5*z = -68) :
  f = -10 ∧ z = 7 := by
  /-
  Given the equations \( f + 3z = 11 \) and \( 3(f - 1) - 5z = -68 \), we need to
  ↪  show that \( f = -10 \) and \( z = 7 \).
  1. Start by isolating \( f \) from the first equation: \( f = 11 - 3z \).
  2. Substitute \( f = 11 - 3z \) into the second equation: \( 3((11 - 3z) - 1) -
  ↪  5z = -68 \).
  3. Simplify the equation step by step:
     - Distribute the 3: \( 3(10 - 3z) - 5z = -68 \).
     - Distribute the 3 again: \( 30 - 9z - 5z = -68 \).
     - Combine like terms: \( 30 - 14z = -68 \).
  4. Solve for \( z \):
     - Subtract 30 from both sides: \( -14z = -98 \).
     - Divide both sides by -14: \( z = 7 \).
  5. Substitute \( z = 7 \) back into \( f = 11 - 3z \):
     - \( f = 11 - 3(7) \).
     - \( f = 11 - 21 \).
     - \( f = -10 \).
  Thus, we have shown that \( f = -10 \) and \( z = 7 \).
  -/
  constructor
  case left =>
    -- Isolate f from the first equation
    have h₂ : f = 11 - 3*z := by linear_combination h₀
    -- Substitute f into the second equation
    have h₃ : 3*(11 - 3*z - 1) - 5*z = -68 := by rw [h₂] at h₁; exact h₁
    -- Simplify the equation to solve for z
    have h₄ : 30 - 14*z = -68 := by linear_combination h₃
    have h₅ : -14*z = -98 := by linear_combination h₄
    have h₆ : z = 7 := by linear_combination h₅ / -14
    exact h₆
  case right =>
    -- Isolate z from the first equation
    have h₂ : z = 7 := by linear_combination h₀
    exact h₂
```&

Let me test it in Lean4

Emmm, it seems the above proof is wrong.

Let me check the error messages.

OK, Here is the error messages:

```bash
line 39

type mismatch
  h₆
has type
  z = 7 : Prop
but is expected to have type
  f = -10 : Prop
```&

```bash
line 42

ring failed, ring expressions not equal
f z : ℂ
h₀ : f + 3 * z = 11
h₁ : 3 * (f - 1) - 5 * z = -68
⊢ 4 + (-(z * 2) - f) = 0
```&

So, I will rethink a Lean4 proof following the steps

   1. Provide the natural language analysis for the theorem based on the Natural
   ↪  language theorem statement, Lean4 theorem statement, my previous proof and
   ↪  the error message.

   2. Draft the Lean4 tactics I should use to solve the problem

   3. Write the output Lean4 code.

Let me analysis the wrong Lean4 solution through the error messages

The first error says:

line 39: type mismatch

$h_6$ has type z = 7 : Prop but is expected to have type f = -10 : Prop

Wait, in the code, I have two cases: left and right. The left case is supposed to
↪  prove f = -10, and the right case z = 7. But in the left case, I derived z =
↪  7 and then tried to use $h_6$ to conclude f = -10, but $h_6$ is z = 7. That's the
↪  type mismatch. So, I need to fix that.

The second error is:

line 42: ring failed, ring expressions not equal

f z : ℂ

```
h₀ : f + 3 * z = 11

h₁ : 3 * (f - 1) - 5 * z = -68

⊢ 4 + (-(z * 2) - f) = 0
```

Hmm, this seems like an unrelated expression that Lean is trying to check. Maybe
↪  there's a mistake in how I'm handling the substitutions or the algebraic
↪  manipulations.

Let me revisit the code. In the left case (proving f = -10), I first derived f =
↪  11 - 3*z from h₀, then substituted that into h₁ to get h₃. Then I simplified
↪  h₃ to get h₄, h₅, and h₆, which led to z = 7. But then I tried to use h₆ to
↪  conclude f = -10, but h₆ is z = 7. I need to instead use z = 7 to find f.

So, in the left case, after getting z = 7, I should substitute that back into f =
↪  11 - 3*z to get f = -10. Then I can use that to conclude f = -10.

Similarly, in the right case, I need to properly handle the substitution and
↪  simplification steps to avoid the ring error.

Let me try adjusting the code accordingly. Instead of having separate cases for
↪  left and right, maybe I should first solve for z, then use that to solve for
↪  f, and then combine the results. Let's see how the corrected code does it.
</Thought>

<Output>
```lean4
import Mathlib
import Aesop

set_option maxHeartbeats 0

open BigOperators Real Nat Topology Rat

/--Given that $f + 3z = 11$ and $3(f - 1) - 5z = -68$, show that $f = -10$ and $z
↪  = 7$.-/
theorem algebra_2varlineareq_fp3zeq11_3tfm1m5zeqn68_feqn10_zeq7
  (f z : ℂ)
  (h₀ : f + 3*z = 11)
  (h₁ : 3*(f - 1) - 5*z = -68) :
  f = -10 ∧ z = 7 := by
  /-
  Given the equations \( f + 3z = 11 \) and \( 3(f - 1) - 5z = -68 \), we aim to
↪  show that \( f = -10 \) and \( z = 7 \).
  1. Start by isolating \( f \) in the first equation: \( f = 11 - 3z \).
  2. Substitute \( f = 11 - 3z \) into the second equation: \( 3(11 - 3z - 1) -
↪  5z = -68 \).
  3. Simplify the equation step by step:
     - Expand the terms inside the parentheses: \( 3(10 - 3z) - 5z = -68 \).
     - Distribute the 3: \( 30 - 9z - 5z = -68 \).
     - Combine like terms: \( 30 - 14z = -68 \).
  4. Solve for \( z \):
```

```
    - Subtract 30 from both sides: \( -14z = -98 \).
    - Divide both sides by -14: \( z = 7 \).
  5. Substitute \( z = 7 \) back into \( f = 11 - 3z \):
    - Calculate \( f \): \( f = 11 - 3(7) = 11 - 21 = -10 \).
  6. Conclude that \( f = -10 \) and \( z = 7 \).
  -/
  -- Isolate f in the first equation
  have h₂ : f = 11 - 3*z := by linear_combination h₀
  -- Substitute f = 11 - 3z into the second equation
  have h₃ : 3*(11 - 3*z - 1) - 5*z = -68 := by rw [h₂] at h₁; exact h₁
  -- Simplify the equation step by step
  have h₄ : 3*10 - 9*z - 5*z = -68 := by linear_combination h₃
  have h₅ : 30 - 14*z = -68 := by linear_combination h₄
  -- Solve for z
  have h₆ : -14*z = -98 := by linear_combination h₅ - 30
  have h₇ : z = 7 := by linear_combination h₆ / -14
  -- Substitute z = 7 back into f = 11 - 3z
  have h₈ : f = -10 := by rw [h₇] at h₂; linear_combination h₂
  -- Conclude the proof
  exact ⟨h₈, h₇⟩
```&
</Output>
```

## F. Training examples

This section, we present the training examples for SFT data and correction data. We can see that in the LoT-TL training stage, we keep the system prompt to switch off the Long CoT thinking. In particular, in the SFT data, we switch off the entire Long CoT., In the correction data training, we only switch off the analysis of error messages in the Long CoT while providing the error message as well as the original proof draft as a component in the Long CoT.

### F.1. LoT-TL Training example for SFT data

```
=== Input Example ===
<|begin_of_sentence|>You are a helpful assistant who will solve every problem
↪  **WITHOUT** Long Chain-of-Thought
### Instruction:
@ Natural language theorem statement:
prover_data_298:
This all simplifies to: $\frac{55*35}{12^7}$
@ Lean4 theorem statement:
```lean4
theorem prover_data_298 :
  ((55 * 35) / (12^7) : ℚ) = (55 * 35) / (12^7)   :=  by
```&
@ Lean4 theorem statement and proof with explanatory comments preceding each
↪  line of code:
### Response:
=== Output Example ===
<Thought>
The user ask not to solve with long CoT, so I will directly write the answer.
</Thought>
<Output>
```lean4
/--This all simplifies to: $\frac{55*35}{12^7}$-/
theorem prover_data_298 :
```

```
  ((55 * 35) / (12^7) : ℚ) = (55 * 35) / (12^7)    :=  by
  /-
  To prove that \(\frac{55 \cdot 35}{12^7}\) is equal to \(\frac{55 \cdot
  ↪  35}{12^7}\), we simply observe that the expressions on both sides of
  ↪  the equality are identical. This identity holds true because the
  ↪  numerator and denominator are the same on both sides.
  -/
  -- We use the fact that the left-hand side and the right-hand side of the
  ↪  equality are the same.
  congr 1
  -- Simplify the power expression.
  <;> simp [pow_succ]
  -- Normalize the numerical values to confirm the equality.
  <;> norm_num
```&
</Output><|end_of_sentence|>
```

## F.2. LoT-TL Training example for correction data

```
=== Input Example ===
<|begin_of_sentence|>You are a helpful assistant who will solve every problem
↪  **WITH** Long Chain-of-Thought. You can make some Lean4 proof draft in
↪  your O1 style Long internal Chain of Thought (CoT) and you have the
↪  ability to test the thing during your thinking. You need to use the error
↪  messages in your Long CoT to guide your proof generation but you are
↪  required **NOT** to provide natural language analysis to the errors.
### Instruction:
@ Natural language theorem statement:
prover_data_2216:
Given non-negative real numbers $a, b, c$ with $a + b + c = 1$. Prove that:
↪  $7(ab + bc + ca) \leq 2 + 9abc$.
@ Lean4 theorem statement:
```lean4
theorem prover_data_2216 (a b c : ℝ) (ha : 0 ≤ a) (hb : 0 ≤ b) (hc : 0 ≤ c)
↪  (habc : a + b + c = 1) : 7 * (a * b + b * c + c * a) ≤ 2 + 9 * a * b * c
↪   :=  by
```&
@ Lean4 theorem statement and proof with explanatory comments preceding each
↪  line of code:
### Response:
<Thought>
Alright, I need to prove the theorem prover_data_2216 using the Lean4 code.
↪  Here is my draft of the proof:
```lean4
theorem prover_data_2216 (a b c : ℝ) (ha : 0 ≤ a) (hb : 0 ≤ b) (hc : 0 ≤ c)
↪  (habc : a + b + c = 1) : 7 * (a * b + b * c + c * a) ≤ 2 + 9 * a * b * c
↪   :=  by
  with aes
```&
Let me test it in Lean4
Emmm, it seems the above proof is wrong.
Let me check the error messages.
OK, Here is the error messages:
```bash
```

```
line 7
unexpected token 'with'; expected '{' or tactic
```&
```bash
line 7
unsolved goals
a b c : ℝ
ha : 0 ≤ a
hb : 0 ≤ b
hc : 0 ≤ c
habc : a + b + c = 1
⊢ 7 * (a * b + b * c + c * a) ≤ 2 + 9 * a * b * c
```&
=== Output Example ===
Since the user ask NOT to provide natural language analysis to the problem.
↪  Let me directly try to write the revised code.
<\Thought>
<Output>
```lean4
/--Given non-negative real numbers $a, b, c$ with $a + b + c = 1$. Prove
↪  that: $7(ab + bc + ca) \leq 2 + 9abc$.-/
theorem prover_data_2216 (a b c : ℝ) (ha : 0 ≤ a) (hb : 0 ≤ b) (hc : 0 ≤ c)
↪  (habc : a + b + c = 1) : 7 * (a * b + b * c + c * a) ≤ 2 + 9 * a * b * c
↪  :=  by
  /-
  Given non-negative real numbers \(a\), \(b\), and \(c\) such that \(a + b +
  ↪  c = 1\), we need to prove that \(7(ab + bc + ca) \leq 2 + 9abc\). The
  ↪  proof involves using algebraic manipulations and inequalities to show
  ↪  that the left-hand side is bounded above by the right-hand side.
  ↪  Specifically, we use the fact that squares of real numbers are
  ↪  non-negative, which helps in establishing the required inequality.
  -/
  -- Normalize the expression involving `a`, `b`, and `c` to prepare for
  ↪  algebraic manipulation.
  ring_nf at habc ⊢
  -- Use `nlinarith` to handle nonlinear arithmetic inequalities. This
  ↪  includes checking non-negativity of expressions involving `a`, `b`, and
  ↪  `c`.
  nlinarith [sq_nonneg (a - 1), sq_nonneg (b - 1), sq_nonneg (c - 1),
    sq_nonneg (a - b), sq_nonneg (b - c), sq_nonneg (c - a),
    sq_nonneg (a + b + c)]
```&
</Output><|end_of_sentence|>
```

# G. Models input example

## G.1. Prover model input-output example

```
### Input Example
IMO problem our prover found
@ Natural language theorem statement:
imo_1961_p1:
(''Hungary'')
Solve the system of equations:


$
\begin{matrix}
\quad x + y + z \\\ &= a \; \, \\
x^2 +y^2+z^2 \\\ &=b^2 \\
\qquad \qquad xy \\\  &= z^2
\end{matrix}
$


where $a $ and $b $ are constants.  Give the conditions that $a $ and $b $
↪   must satisfy so that $x, y, z $ (the solutions of the system) are
↪   distinct positive numbers.

@ Lean4 theorem statement:
```lean4
theorem imo_1961_p1 (x y z a b : ℝ) (h₀ : 0 < x ∧ 0 < y ∧ 0 < z) (h₁ : x ≠ y)
↪   (h₂ : y ≠ z)
  (h₃ : z ≠ x) (h₄ : x + y + z = a) (h₅ : x ^ 2 + y ^ 2 + z ^ 2 = b ^ 2) (h₆ :
    ↪   x * y = z ^ 2) :
  0 < a ∧ b ^ 2 < a ^ 2 ∧ a ^ 2 < 3 * b ^ 2 := by
```&

@ Lean4 theorem statement and proof with explanatory comments preceding each
↪   line of code:
 Response:
<Thought>
Alright, I should do the following:
  1. Provide the natural language analysis for the theorem based on the
   ↪   Natural language theorem statement.
  2. Draft the Lean4 tactics I should use to solve the problem
  3. Write the output Lean4 code.
The user also asks that I should avoid using the keyword `sorry` to give up
↪   the proof, so I will not write it in my Lean4 code.
### Output Example
The `imo_1961_p1` can be proofed by analyzing the given equations and
↪   deriving necessary inequalities to ensure that the solutions are distinct
↪   positive numbers.
</Thought>
<Output>
```lean4
theorem imo_1961_p1 (x y z a b : ℝ) (h₀ : 0 < x ∧ 0 < y ∧ 0 < z) (h₁ : x ≠ y)
↪   (h₂ : y ≠ z)
```

```
  (h₃ : z ≠ x) (h₄ : x + y + z = a) (h₅ : x ^ 2 + y ^ 2 + z ^ 2 = b ^ 2) (h₆ :
  ↦  x * y = z ^ 2) :
  0 < a ∧ b ^ 2 < a ^ 2 ∧ a ^ 2 < 3 * b ^ 2 := by
  -- We need to prove three inequalities: 0 < a, b^2 < a^2, and a^2 < 3 *
  ↦  b^2.
  refine' ⟨_, _, _⟩
  -- First, prove 0 < a.
  -- Since x, y, z are positive, their sum a must also be positive.
  . nlinarith [h₀.1, h₀.2.1, h₀.2.2, sq_nonneg (x - y), sq_nonneg (y - z),
  ↦  sq_nonneg (z - x)]
  -- Next, prove b^2 < a^2.
  -- Using the given equations and inequalities, derive the necessary
  ↦  inequalities.
  . nlinarith [mul_self_pos.mpr (sub_ne_zero.mpr h₁), mul_self_pos.mpr
  ↦  (sub_ne_zero.mpr h₂),
      mul_self_pos.mpr (sub_ne_zero.mpr h₃), sq_nonneg (x - y), sq_nonneg (y -
      ↦  z), sq_nonneg (z - x)]
  -- Finally, prove a^2 < 3 * b^2.
  -- Combine all the given information to derive this inequality.
  . nlinarith [mul_self_pos.mpr (sub_ne_zero.mpr h₁), mul_self_pos.mpr
  ↦  (sub_ne_zero.mpr h₂),
      mul_self_pos.mpr (sub_ne_zero.mpr h₃), sq_nonneg (x - y), sq_nonneg (y -
      ↦  z), sq_nonneg (z - x)]
```&
</Output>
```

## G.2. Corrector model input-output example

```
### Input Example
You are a helpful mathematical assistant specialized in formal theorem
↦  proving using Lean4.
Your objectives:
1. Read and interpret the Lean4 theorem statement and any error messages.
2. **If a previous proof attempt was incorrect, analyze its exact mistakes
↦  and completely discard or rewrite the proof as needed.**
3. **Avoid reusing incorrect proof structures or strategies unless explicitly
↦  validated as correct.**
4. **Address all error messages** by modifying the proof structure as needed.
5. Provide a detailed thought process in the <Thought> section, but **only
↦  place the corrected Lean4 code in the <Output> section**.
6. **Ensure the new proof is logically valid and does not use `sorry`.**
↦  Instruction:Below are some **correct Lean4 theorem proof examples** for
↦  your reference. Use them as guidance when constructing the revised proof.
↦  Ensure that your final proof aligns with these examples in terms of
↦  clarity, structure, and correctness. Instruction: @ Natural language
↦  theorem statement:
mathd_algebra_392
The sum of the squares of three consecutive positive even numbers is $12296$.
↦  Find the product of the three numbers divided by $8$. Show that it is
↦  32736.

@ Lean4 theorem statement:
```lean4
theorem mathd_algebra_392
```
```

```
  (n : ℕ)
  (h₀ : Even n)
  (h₁ : ((n:ℤ) - 2)^2 + (n:ℤ)^2 + ((n:ℤ) + 2)^2 = 12296) :
  ((n - 2) * n * (n + 2)) / 8 = 32736 :=
```&

@ Lean4 theorem statement and proof with explanatory comments preceding each
↪  line of code:
 Response:
<Thought>
Alright, I need to prove the theorem prover_data_68493 using the Lean4 code.
↪  Here is my draft of the proof:

```lean4
theorem mathd_algebra_392
  (n : ℕ)
  (h₀ : Even n)
  (h₁ : ((n:ℤ) - 2)^2 + (n:ℤ)^2 + ((n:ℤ) + 2)^2 = 12296) :
  ((n - 2) * n * (n + 2)) / 8 = 32736 := by
  -- Convert the equation to integers for easier manipulation
  norm_cast at h₁
  -- Expand and simplify the given equation
  ring_nf at h₁
  -- Solve for n using numerical reasoning
  have h₂ : n = 24 := by
    nlinarith
  -- Substitute n = 24 into the expression and simplify
  norm_num [h₂]
```&
Let me test it in Lean4
Emmm, it seems the above proof is wrong.
Let me check the error messages.
OK, Here is the error messages:
```bash
line 18

linarith failed to find a contradiction
case h1.h
n : ℕ
h₀ : Even n
h₁ : Int.subNatNat n 2 ^ 2 + ↑(n ^ 2) + ↑(4 + n * 4 + n ^ 2) = 12296
a† : n < 24
⊢ False
failed
```&
```bash
line 11

unsolved goals
n : ℕ
h₀ : Even n
h₁ : Int.subNatNat n 2 ^ 2 + ↑(n ^ 2) + ↑(4 + n * 4 + n ^ 2) = 12296
h₂ : n = 24
⊢ False
```

```
```&
```

So, I will rethink a Lean4 proof following the steps
  1. Provide the natural language analysis for the theorem based on the
  ↪  Natural language theorem statement, Lean4 theorem statement, my
  ↪  previous proof and the error message.
  2. Draft the Lean4 tactics I should use to solve the problem
  3. Write the output Lean4 code.
Let me analysis the wrong Lean4 solution through the error messages.
### Output Example
From the error messages, it seems the following points:
- `linarith failed to find a contradiction` suggests that the linear
  ↪  arithmetic tactic was unable to derive a contradiction from the given
  ↪  hypotheses, indicating that the assumptions might not lead to a
  ↪  contradiction as expected.
- `case h1.h` indicates a specific case within a larger context where a
  ↪  contradiction was expected but was not found, suggesting that the
  ↪  assumption `n < 24` might not be valid under certain conditions.
- `unsolved goals` means that after applying the tactics, there are still
  ↪  unresolved goals left, indicating that the tactics applied were
  ↪  insufficient to complete the proof.
From these error messages, I realize that my previous approach might not have
  ↪  handled the specifics of the problem correctly. Therefore, I need to
  ↪  revise my proof strategy.
First, I will analyze the natural language theorem statement and the provided
  ↪  Lean4 theorem statement to understand the problem fully.
The problem involves three consecutive even numbers whose squares sum up to
  ↪  12296. The goal is to find the product of these three numbers divided by
  ↪  8 and show that it equals 32736.
Given that the numbers are consecutive even numbers, I can denote the middle
  ↪  number as `n`. Then, the three consecutive even numbers can be
  ↪  represented as `n-2`, `n`, and `n+2`.
The sum of their squares is given by:
```
((n - 2)^2) + (n^2) + ((n + 2)^2) = 12296
```

Expanding these squares, we get:
```
(n^2 - 4n + 4) + n^2 + (n^2 + 4n + 4) = 12296
```

Combining like terms:
```
3n^2 + 8 = 12296
```

Subtracting 8 from both sides:
```
3n^2 = 12288
```

Dividing both sides by 3:
```
n^2 = 4096
```

Taking the square root of both sides:
```

```
n = 64
```

Since we are dealing with squares, we take the positive root because we are
↪ dealing with natural numbers.
Now, the three consecutive even numbers are:
```

n – 2 = 62
n = 64
n + 2 = 66
```

The product of these numbers is:
```

62 * 64 * 66
```

To compute this product, I can break it down into smaller factors:
```

62 * 64 = 3968
3968 * 66 = 3968 * (60 + 6) = 3968 * 60 + 3968 * 6 = 238080 + 23808 = 261888
```

Now, divide this product by 8:
```

261888 / 8 = 32736
```

Thus, the final result matches the expected outcome.
Now, I need to translate this reasoning into Lean4 code.
I will follow these steps:
1. Define the theorem with the given parameters.
2. Convert the given integer expressions to natural numbers.
3. Expand and simplify the equation using algebraic manipulation.
4. Solve for `n` using numerical methods.
5. Compute the product of the three numbers and verify that dividing by 8
↪ gives the desired result.
I will also include necessary imports and use appropriate tactics to handle
↪ the calculations.
Now, I will proceed to write the Lean4 code following this plan.
</Thought>

<Output>
```lean4
theorem mathd_algebra_392
  (n : ℕ)
  (h₀ : Even n)
  (h₁ : ((n:ℤ) – 2)^2 + (n:ℤ)^2 + ((n:ℤ) + 2)^2 = 12296) :
  ((n – 2) * n * (n + 2)) / 8 = 32736 := by
  -- Convert the equation to integers for easier manipulation
  push_cast at h₁
  -- Expand and simplify the given equation
  ring_nf at h₁
  -- Solve for n using numerical reasoning
  have h₂ : n = 64 := by
    nlinarith
  -- Substitute n = 64 into the expression and simplify
  subst h₂
  norm_num
```

```
```&
</Output>
```

