# OpenReview forum: "MA-LoT: Model-Collaboration Lean-based Long Chain-of-Thought Reasoning enhances Formal Theorem Proving"
_ICML.cc/2025/Conference — ICML 2025 poster_

### Official Review · Reviewer_tVG8 · 2025-03-10

**Overall Recommendation:** 3

**Summary:**

This paper proposes the MA-LoT framework, which incorporates a long CoT with an iterative refinement approach to improve formal theorem-proving ability. The model first makes an attempt using NL planning followed by FL proof. If the attempt is incorrect, the initial prompt is combined with the error message to generate a second attempt. To enable this capability, the model is first fine-tuned on NL long CoT data from the OpenO1-SFT-Pro dataset. Then, it is trained on FL proof data with NL annotations. To collect FL-NL aligned data, this paper also introduces a pipeline for dataset creation.

**Claims And Evidence:**

1. The model claims to be the first to use a multi-agent framework for Lean4 theorem proving. However, this framework essentially involves the model making an initial attempt, receiving an error message, and then generating a second attempt based on the first attempt and the error message. This process is repeated iteratively. A similar approach has already been explored. For example, DeepSeek-Prover-v1.5 has utilized error messages in a tree-based search setting. Additionally, many other papers, such as “Proof Automation with Large Language Models” and “An In-Context Learning Agent for Formal Theorem Proving”, have investigated similar ideas.
2. DeepSeek-Prover-v1.5-RL achieved an accuracy of 51.6% ± 0.5% on the miniF2F-test for 128@pass whole-proof generation in the original paper, whereas this paper reports the accuracy as 48.36%. The whole-proof accuracy of LoT is 52.05%, which only marginally surpasses the original DeepSeek-Prover-v1.5-RL results (within one standard deviation). For tree search-based methods, DeepSeek-Prover-v1.5-RL achieves an accuracy as high as 63.5%, which exceeds MA-LoT’s performance. These experimental results suggest that MA-LoT does not outperform the SOTA methods.

**Essential References Not Discussed:**

N/A

**Experimental Designs Or Analyses:**

The overall experimental design is good. However, it would be beneficial to include computational efficiency in the evaluation; otherwise, the tree-search-based DeepSeek-Prover-v1.5-RL may achieve higher accuracy. Additionally, the high computational overhead is a major drawback of long CoT reasoning.

**Methods And Evaluation Criteria:**

The metric and evaluation criteria are fine.

**Other Comments Or Suggestions:**

1. I will currently rate the paper as a 2, but I am open to increasing the score if my concerns are properly addressed. Overall, I believe the paper is suitable for ICML; however, the unnecessary emphasis on the multi-agent framework (is it truly essential to make it the core of the story?) and the omission of some SOTA results raise concerns. In my opinion, the main contribution is extending the long CoT method to the formal theorem-proving method.
2. Please remember to include a short title.

**Other Strengths And Weaknesses:**

N/A

**Questions For Authors:**

N/A.

**Relation To Broader Scientific Literature:**

This paper extends the long CoT method to the formal theorem proving.

**Theoretical Claims:**

N/A.

---

> ### Author Rebuttal · Authors · 2025-03-31
>
> Dear Reviewer tVG8
>
> Thank you so much for your appreciation of our work. We are sincerely grateful that you consider it suitable for ICML.
>
> With our deepest thanks for your constructive comments, we would like to share the latest results on the MiniF2F-Test dataset using a new base model named **Goedel-Prover**:
> |Method|Budget|MiniF2F-Test|
> |-|-|-|
> |Goedel-Prover-SFT|pass@32|55.33%|
> |MA-LoT (Goedel)|16+8×2|**61.07%**|
>
> The result under the pass@32 metric demonstrates the SOTA performance of our method. The improvement also highlights the versatility of the MA-LoT framework.
>
> To address your concerns more clearly, we summarize your points and provide our responses below:
>
> 1. **Multi-agent setup problem:** We are truly thankful for your detailed understanding of our work and related efforts in the field. You raised an important point that some prior works may have used error messages in conjunction with tree search. However, to our knowledge, neither DeepSeek-Prover-v1.5 nor Lean-STaR uses error messages during the tree search process. Our work, as far as we know, is the first in the context of Lean4 theorem proving to explicitly incorporate error messages within a multi-agent Long CoT framework. We will clarify this point in the next version.
>
>     The motivation for our multi-agent setup is to **separate cognitive tasks**—with the prover agent handling high-level proof planning and the corrector agent focusing on fine-grained error correction. This is more effective than simply using a prover agent for tree search without such separation.
>
>     To validate this, we conducted an experiment where the prover agent was used to perform both full-proof generation and error correction. This was done by including the draft code and error message as comments in the theorem statement. The results using Goedel-Prover on the MiniF2F-Test are as follows:
>     |Methods|Prover|Round 1|Round 2|
>     |-|-|-|-|
>     |Prover-correction|54.92%|56.15%|57.38%|
>     |MA-LoT|54.92%|59.43%|61.07%|
>
>     As shown, using the prover for error correction results in suboptimal performance. However, we appreciate your suggestion and are happy to tone down the "first multi-agent" claim. We are also open to renaming it as **“Prover-Corrector Collaboration”** or adopting any alternative phrasing that better conveys the contribution without overstating the novelty.
>
> 2. **Concerns about SOTA performance:** We are grateful for your thoughtful critique and would like to break down our response into three parts:
>     1. Differences in measured baselines vs. reported values: We are aware that our measured baseline results differ from those reported in the DeepSeek-Prover paper. This discrepancy likely stems from our machine being unable to install vllm. A similar degradation was also observed in the Goedel-Prover baseline and other experiments. We will clarify this technical limitation in the paper.
>     2. Suboptimal whole-proof accuracy: We appreciate your effort to compare the model capability of LoT-Solver to other base models. While our results may be degraded due to infrastructure constraints, our focus is on the overall framework rather than standalone model performance. Even with baseline degradation, the MA-LoT framework still achieves a 5.64% relative improvement over DeepSeek-Prover-v1.5 and 6.02% over Goedel-Prover. Additionally, we observed some theorems that cannot be solved even with a large computational budget (e.g., pass@256). We include a detailed example in https://anonymous.4open.science/r/MA-LoT_Rebuttal_repo-516F
>
>     3. Other works have higher accuracy under a larger computational budget: We appreciate the suggestion to explore higher performance using larger budgets (e.g., 32 × 6,400 search). However, the main contribution of our work lies in the methodology. Therefore, we only conducted experiments under a smaller budget. However, we are more than happy to include large-budget experiments in a future version if you think it necessary.
> 3. **Computational efficiency for the Long CoT method:** We appreciate your suggestion that Long CoT may consume more computational resources under the same pass rate. To address this, we evaluated GPU usage and compared the efficiency of MA-LoT against baselines.
>
>     We found that MA-LoT consumes ~1.7× more GPU hours. For fairness, we scaled the sampling budget for DS-Prover-v1.5-RL and Goedel-Prover accordingly and compared them with MA-LoT. As shown below, MA-LoT still outperforms the baselines, demonstrating a more efficient use of the computational budget:
>     ||Budget|MiniF2F-Test|
>     |-|-|-|
>     |DS-Prover-v1.5-RL|pass@217|52.87%|
>     |MA-LoT (DS)|64+32×2|54.51%|
>     |Goedel-Prover|pass@53|58.20%|
>     |MA-LoT (Goedel)|16+2×8|**61.07%**|
>
> 4. **Title concerns:** We will use a shorter title in the next version.
>
> We are grateful for the reviewer’s appreciation of our methodology and constructive comment, and we hope that our rebuttal has settled your concerns.
>
> Sincerely,
>
> 9122 Author team

---

> > ### Comment · Reviewer_tVG8 · 2025-04-03
> >
> > Thanks for your response.
> >
> > 1. Sorry for the earlier mistake about Deepseek-prover-v1.5 using error messages. It seems they do not use them but rather truncate at the first verification error. However, using error messages to improve a second attempt is not a new idea in formal math proving. A quick search led me to this paper: https://arxiv.org/pdf/2309.15806. That said, I am glad to see that you would like to tone down the "first multi-agent" claim.
> > 2. I think it would be appropriate to notify the readers that the baseline result you reported is different from the original paper.
> > 3. In the discussion on efficiency, a more suitable metric might be the number of new tokens generated, since GPU hours can be affected by several external factors. I hope this comparison will be included in your revision.
> >
> > I will increase the score by 1 if you can confirm that these changes will be reflected in the updated version.

---

> > > ### Author Response · Authors · 2025-04-04
> > >
> > > Dear reviewer tVG8,
> > >
> > > Thank you so much for your response. We confirm the following change will be included in the next version of our paper.
> > >
> > > 1. We will tone down our "first multi-agent" claim and perform. We also appreciate you highlighting the prior work (arXiv:2309.15806) on using error messages to guide retries. We will cite this paper to properly acknowledge the existing literature and clarify how our method differs.
> > >
> > > 2. Regarding the baseline discrepancy, we agree it is important to maintain transparency. We will explicitly notify the authors of the difference between the reproduced result and the original result in the updated version.
> > >
> > > 3. We agree that GPU hours are influenced by many external factors and may not be the most reliable metric for efficiency. The experiment of comparison between generated tokens will be provided in the upcoming version of the paper.
> > >
> > > We confirm these changes will all be reflected in the updated version. Thank you again for your constructive and helpful review.
> > >
> > > Best,
> > >
> > > 9122 Author team

---

### Official Review · Reviewer_bbL4 · 2025-03-12

**Overall Recommendation:** 2

**Summary:**

This paper introduces MA-LoT, a multi-agent framework for theorem proving in Lean 4, integrating natural language (NL) reasoning with formal language (FL) verification via Long Chain-of-Thought (Long CoT). Using a novel LoT-Transfer Learning pipeline, MA-LoT enhances proof coherence and depth, outperforming GPT-4 (22.95%), single-agent tree search (49.18%), and whole-proof generation (48.30%) on the Lean4 MiniF2F-Test dataset, achieving 54.92% accuracy. Results highlight the potential of structured NL-FL reasoning for more effective formal proof generation.

**Claims And Evidence:**

While the paper claims to introduce “the first multi-agent framework for Lean 4 theorem proving that balances NL reasoning and FL verification in Long CoT”, the necessity of a multi-agent setup is not convincingly justified. The interactions between agents could potentially be replicated by a single agent operating sequentially, generating a proof first and then refining it according to the feedback from Lean in an iterative manner.

The paper does not provide compelling evidence that a multi-agent approach yields inherent advantages over a well-structured single-agent framework following the same reasoning pipeline. Without empirical ablations comparing multi-agent interaction to sequential single-agent processing, the claim that multi-agent coordination is essential remains unsubstantiated.

However, the Long CoT aspect of the claim is well-supported, as the paper presents new training data and field-specific alignment strategies that demonstrate its impact on proof generation quality.

**Essential References Not Discussed:**

The references look reasonable.

**Experimental Designs Or Analyses:**

The experimental designs and analyses look reasonable. Limiting other approaches sample budget is acceptable for fair comparison. It is known, however, that some of the methods (e.g., InternLM2.5-StepProver (7B) and DeepSeek-Prover-V1.5-RL (7B) ) in table 1 surpass the best performance of the submission when given more sample budget.

**Methods And Evaluation Criteria:**

The proposed methods make sense for formal theorem proving. My impression is that the Long CoT pipeline including the field-specific alignment strategy is perhaps what contributes to the improved performance most.

**Other Comments Or Suggestions:**

In the abstract and conclusion the authors report 54.92% accuracy, but throughout the entire experiment section I only see 54.51%. A typo?

**Other Strengths And Weaknesses:**

My major complaint is the advertisement of the approach being multi-agent. The framework can in principle be done by a single agent operating sequentially. Moreover, Table 1 suggests that the major performance boost comes from the long CoT, and the benefits from correction based on Lean’s feedback seem relatively small.

Factoring out the contribution of the so-claimed “multi-agent”, the long CoT pipeline itself is not particularly significant for a research paper. I do appreciate the amount of work put into compiling / enhancing a new dataset for Lean, though.

**Questions For Authors:**

Nothing in particular.

**Relation To Broader Scientific Literature:**

The long CoT pipeline can potentially be used to generate training data suitable for theorem provers other than Lean.

**Theoretical Claims:**

The paper is mostly empirical.

---

> ### Author Rebuttal · Authors · 2025-03-31
>
> Dear Reviewer bbL4
>
> Thank you so much for your valuable comments on our paper and your appreciation of our contribution to both dataset construction and training methodology. Your encouragement truly motivates us to continue pursuing research in this field.
>
> Because formal theorem proving is a fast-evolving field, we would like to share with you the latest results on the MiniF2F-Test dataset for our model using a new base model named **Goedel-Prover.**
>
> | Method | Budget | MiniF2F-Test |
> | --- | --- | --- |
> | Goedel-Prover-SFT | pass@32 | 55.33% |
> | MA-LoT (Goedel) | 16+8×2 | **61.07%** |
>
> The result under pass@32 demonstrates the SOTA performance of our method. The improvement also highlights the versatility of the MA-LoT framework.
>
> To better address your concerns, we have summarized your key questions and provided our responses below:
>
> 1. **The necessity of multi-agent setup:** We understand your concern regarding the multi-agent setup. We acknowledge that our approach differs from traditional graph-based multi-agent systems that involve complex interaction between many agents. Instead, our setup sequentially applies two agents.
>
>     The motivation behind this design is to **separate cognitive tasks**: high-level proof planning is handled by the **prover agent**, and fine-grained error correction is handled by the **corrector agent**. Our focus lies in the division of labor via the multi-agent framework, rather than building a complex interaction graph across multiple LLMs.
>
>     To further address your concern, we conducted an experiment using a **single prover agent** to sequentially perform both whole-proof writing and error correction. In this setup, the LoT-Solver model based on the Goedel-Prover was prompted to act solely as a prover, and the error correction was done by inserting the draft code and error messages as comments in the theorem statement. The results are as follows:
>
>     | Methods | Prover | Round 1 | Round 2 |
>     | --- | --- | --- | --- |
>     | Prover-correction | 54.92% | 56.15% | 57.38% |
>     | MA-LoT | 54.92% | 59.43% | 61.07% |
>
>     We find that using a single prover agent for correction yields suboptimal results. The model often repeats the original proof without deeply analyzing the error, whereas the corrector agent’s **Long CoT thinking** encourages more effective debugging and correction.
>
> 2. **Advertis**e**ment of the multi-agent:** We greatly appreciate your suggestion regarding the potential over-emphasis on the multi-agent setup. While we do believe that separating the prover and corrector roles is important, we are happy to **reframe the terminology** in a future version of the paper. If necessary, we will tone down the “first multi-agent” claim or rename it as **“prover-corrector collaboration”**. Additionally, we are also open to other suggestions regarding the naming and paper writing.
> 3. **Computational budget concern:** You raised an important point about whether improved results could be achieved with a larger sampling budget for DeepSeek-Prover-v1.5 and InternLM-Step-Prover. We would like to clarify that, unlike these works (and Goedel-Prover), our study focuses on **methodology**—offering a general framework that can be applied to **any base model**.
>
>     Our experiments demonstrate that applying MA-LoT to multiple base models results in significant performance improvements. We also plan to **open-source** the dataset and training code in the camera-ready version.
>
>     Given the methodological focus and computational cost constraints, we chose not to perform extremely large-scale experiments (e.g., pass@25,600) in this version. However, we are happy to include such experiments in a future version if the reviewer deems them necessary.
>
> 4. **Long CoT is not significant enough:** Thank you for recognizing our **Long CoT training and inference framework**. We would like to clarify that our work is not simply about applying Long CoT to formal theorem proving. Instead, our motivation lies in using **formal guidance** as the backbone of Long CoT to support more comprehensive reasoning.
>
>     The novelty of our approach is in the deep integration of natural language (NL) reasoning and formal language (FL) feedback, where formal language acts as a hidden regularizer to guide the model’s behavior. Therefore, the significance of Long CoT is not only in the method itself but also in the conceptual framework it introduces. Reviewer tBG8 also considers our contribution is suitable for ICML. We will add this clarification in the next version of the paper.
>
> 5. **Typo problem:** Yes, the “54.92%” figure in the abstract and conclusion is a typo. We will update it to reflect the correct result based on the Goedel-Prover base model in the next version.
>
> We are more than grateful for the reviewer’s appreciation of our Long CoT training inference framework and hope that our rebuttal has settled your concerns.
>
> Sincerely,
>
> 9122 Author team

---

### Official Review · Reviewer_Lfxq · 2025-03-14

**Overall Recommendation:** 3

**Summary:**

This paper introduces MA-LoT, a multi-agent framework for formal theorem proving in Lean 4 combining natural language reasoning with verifier feedback. MA-LoT employs two "agents" (same LLM prompted in different ways): a prover that generates proofs using "Long" Chain-of-Thought reasoning, and a "corrector" that sees feedback from Lean and tries to repair proofs. The authors develop a training pipeline (LoT-TL) to gather long CoT examples from problems in existing datasets. Experiments on minif2f show MA-LoT generally outperforms other methods.

**Claims And Evidence:**

There are some vague claims that I feel the authors could clarify either what exactly they mean, or simply rephrase to remove them.

- "Long CoT": what exactly does "Long CoT" mean in comparison to plain old "CoT"? I know there is some intuition of what this means by looking at the examples from OpenAI o1 that were released in OpenAI's blog post. The paper here ablates Long CoT by switching CoT entirely. Since "Long CoT" is even in the title, and the motivation mentions it, I think it's important to either make this more precise or to simply call this chain-of-thought. L197, for instance, claims that the prompt specifically turns on "Long CoT", but I don't think the examples in the Appendix are particularly "Long", or that this wouldn't happen with more standard CoT prompting. One option here could be to show that, during training, the content of the <Thought> part of the output is indeed getting steadily longer.
- The claim to be the first multi-agent setup for theorem proving is either too specific (if you just mean "for Lean 4"), or innacurate, since Baldur [1] showed a very similar setup a few years ago (but in Isabelle/HOL).
- Some qualitative claims about how the proofs look ("insightful", "coherent", etc) are not evaluated in any systematic way, just mentioned in the text.

[1] FIRST, Emily et al. Baldur: Whole-proof generation and repair with large language models. In: Proceedings of the 31st ACM Joint European Software Engineering Conference and Symposium on the Foundations of Software Engineering. 2023. p. 1229-1241.

**Essential References Not Discussed:**

Baldur (FSE '23), mentioned several times above.

**Experimental Designs Or Analyses:**

The minif2f experiment seems mostly standard. It's hard to know if the compute budget is fairly matched, since I think the approach proposed here likely generates a lot more tokens per call (the authors match the number of calls, pass@128 vs 2 x 64). This should be discussed.

**Methods And Evaluation Criteria:**

The authors only evaluate on minif2f. While it is still a challenging benchmark, it's unclear if the resulting model performs better in other distributions of theorems, like in the mathlib splits in LeanDojo, or on Lean Workbook itself.

The claim to outperform GPT-4, but comparing to GPT-4-Turbo, is also perhaps unwarranted: this is a relatively old model by now, and trained much before the base model the authors used (DeepSeek-Prover) was trained.

**Other Comments Or Suggestions:**

* [3.2] "for training for training" (repeated)
* Table 2: column name should not be "Method"
* Table 3: witch-off -> switch off?

**Other Strengths And Weaknesses:**

N/A

**Questions For Authors:**

1. Have you measured the actual token generation differences between MA-LoT and baselines? Since each of your calls is likely to be much more expensive, tokens generated would likely be a fairer metric to match compute budgets. As it stands, it's unclear whether the gap between LoT and MA-LoT will remain, for instance.
2. One feature of "Long CoT" in o1/s1/r1 seems to be that increased thinking time improves performance. Is that also true for MA-LoT? This could perhaps help you characterize your approach as "Long CoT". Otherwise, it doesn't look different from standard CoT, even though "Long" was emphasized several times.
3. "The corrector agent functions like the tree-search method." -> in what sense? This does not read like tree search.

**Relation To Broader Scientific Literature:**

The paper tackles minif2f, which is still a challenging benchmark. The idea to have "two agents" is essentially the framework in Baldur, of doing proof repair besides proof generation (modulo details and formal language).

**Theoretical Claims:**

N/A

---

> ### Author Rebuttal · Authors · 2025-03-30
>
> Dear Lfxq,
>
> We would like to offer our sincere thanks for your constructive comments and valuable suggestions, which have helped make our paper more coherent.
>
> Because formal theorem proving is a fast-evolving field, we would like to share with you the latest results on the MiniF2F-Test dataset for our model using a new base model named **Goedel-Prover**.
> |Method|Budget|MiniF2F-Test|
> |-|-|-|
> |Goedel-Prover-SFT|pass@32| 55.33%|
> |MA-LoT (Goedel)|16+8×2|**61.07%**|
>
> The result under pass@32 demonstrates the SOTA performance of our method. The improvement also highlights the versatility of the MA-LoT framework.
>
> Below are our responses to your comments:
> 1. **Difference between Long CoT and normal CoT:** We have included a comparison between Long CoT and normal CoT in Appendix A.5. Generally speaking, the key difference lies in the *multi-round rethinking* capability of Long CoT, which naturally fits the context of formal theorem proving. In practice, Long CoT typically involves several rounds of natural language reasoning before producing the final output, leading to more thoroughly-thought results. In some cases, the LLM even analyzes tactics in detail before generating code under Long CoT.
> 2. **Concerns about multi-agent setup:** We understand the reviewer may have seen related work that uses error messages for proof correction. However, we would like to clarify our motivation for the multi-agent setup. Our goal is to **separate cognitive tasks**: high-level proof planning is handled by the prover agent, while fine-grained error correction is handled by the corrector agent.
>
>     Our work differs from Baldur in that we incorporate natural language reasoning and multi-round thinking in Long CoT. If necessary, we are happy to tone down the “first multi-agent” claim, restrict it to the Lean4 context, rename it as “prover-corrector collaboration”, or any other suggestions regarding the naming.
>
>     Additionally, we emphasize the benefit of separating the prover and corrector agents. We designed an experiment where the prover agent attempts to correct errors using draft code and error messages as inline comments. The results are as follows, from the results, we can see that the additional corrector agent is vital for better performance.
>     |Methods|Prover|Round 1|Round 2|
>     |-|-|-|-|
>     |Prover-correction |54.92%|56.15%|57.38%|
>     |MA-LoT|54.92%|59.43%|61.07%|
> 3. **Qualitative study not supported:** We do provide concrete support for the qualitative claims in the paper’s case study. To further address your concern, we have added more examples in the anonymous Git https://anonymous.4open.science/r/MA-LoT_Rebuttal_repo-516F
> 4. **Additional benchmarks beyond MiniF2F:** We conducted experiments on the ProofNet-Test using the Goedel-Prover. The relative improvement of 27.33%, detailed results are shown below:
>     ||Budget|ProofNet-Test|
>     |-|-|-|
>     |**Goedel-Prover**|Pass@32|12.15%|
>     |**MA-LoT (Goedel)**|16+16|**15.47%**|
> 5. **GPT-4 may be an old baseline:** We have updated our baselines for both closed-source and open-source models, now including DeepSeek-V3 and R1-Distilled-Qwen-7B. The MiniF2F-Test results are as follows, showing that MA-LoT still outperforms all the methods:
>     ||**Budget**|**MiniF2F-Test**|
>     |-|-|-|
>     |DeepSeek-V3|pass@32|33.61%|
>     |R1-Distilled-7B|pass@32|51.23|
>     | MA-LoT (Goedel)|16+2×8|**61.07%**|
> 6. **Computation budget for Long CoT:** In our paper, we control the number of proofs written by the LLMs rather than the number of tokens generated, as Lean-STaR also includes natural language analysis but does not use tokens as a comparison metric.
>
>     However, we consider your review meaningful so we conduct additional experiments based on aligned GPU times. Our method consumes approximately **1.7× more GPU-hours** than traditional CoT. For fairness, we increased the sampling budget by the same factor for both DeepSeek-Prover-v1.5-RL and Goedel-Prover, and compared them with the MA-LoT framework. As shown below, even with increased computational budget, MA-LoT still outperforms the baselines, indicating a more efficient use of resources:
>     | |Budget|MiniF2F-Test|
>     |-|-|-|
>     |DS-Prover-v1.5-RL |pass@217| 52.87%|
>     |MA-LoT (DS)| 64+32×2|54.51%|
>     |Goedel-Prover |pass@53|58.20%|
>     |MA-LoT (Goedel) |16+2×8|**61.07%**|
>
>     Qualitatively, we have observed some theorems that cannot be proved using even high-budget base models; details are demonstrated in the git repo provided above.
> 7. **Corrector like tree-search:** Our statement was meant as a **conceptual analogy**, not a literal claim. We will revise or remove it in the next version to avoid confusion and improve clarity.
> 8. **Typos:** Thank you for pointing these out. We will correct them and fix all other typos in the next version.
>
> We hope this rebuttal addresses your concerns, and we are deeply grateful for your insightful suggestions, which have helped improve the clarity of our paper.
>
> Best,
>
> 9122 Author team

---

> > ### Comment · Reviewer_Lfxq · 2025-04-06
> >
> > I thank the authors for engaging with my concerns, which have been largely addressed. Assuming the promised revisions to the paper (e.g., on the multi-agent framing, etc), as well as the inclusion of the new results, I have revised my score.

---

> > > ### Author Response · Authors · 2025-04-07
> > >
> > > Thanks a lot for your appreciation of our research, we will certainly revise the paper by toning down the "multi-agent" claim and including all the new results provided in the rebuttal period. Thank you again for the revision of score!

---

### Decision · Program_Chairs · 2025-05-01

**Decision:**

Accept (poster)

**Comment:**

The paper introduces the MA-LoT framework, a multi-agent system designed to enhance formal theorem proving in Lean4 by integrating natural language reasoning with formal language verification through a Long Chain-of-Thought (Long CoT) approach. The authors claim that this framework is the first of its kind for Lean4 theorem proving, achieving significant improvements over existing methods.

(+) The framework demonstrates a notable improvement in accuracy on the Lean4 MiniF2F-Test dataset, outperforming several baseline models, including GPT-4 and other single-agent methods.
(+) The authors provide extensive experimental results, including comparisons with various baselines and ablation studies to highlight the effectiveness of their approach.
(-) The necessity and novelty of the multi-agent setup were questioned by reviewers. The interactions between agents could potentially be replicated by a single agent operating sequentially, and similar approaches have been explored in other contexts.
(-) The high computational overhead of the Long CoT reasoning was noted as a drawback, and the evaluation did not initially include a suitable metric for computational efficiency.

The authors actively engaged with reviewers during the rebuttal phase, addressing concerns and providing additional experiments and clarifications.

The paper presents a promising approach to formal theorem proving with a well-structured framework and significant empirical results. While there are concerns about the novelty of the multi-agent setup and baseline comparisons, the authors have addressed these issues satisfactorily in their rebuttal. Assuming the promised revisions are made, I recommend the paper for acceptance.